# Stochastic Self-Organization in Multi-Agent Systems

**Nurbek Tastan**[1]    **Samuel Horváth**[1]    **Karthik Nandakumar**[1,2]
[1]Mohamed bin Zayed University of Artificial Intelligence (MBZUAI), UAE
[2]Michigan State University (MSU), USA
{nurbek.tastan,samuel.horvath}@mbzuai.ac.ae, nandakum@msu.edu

## Abstract

Multi-agent systems (MAS) based on Large Language Models (LLMs) have the potential to solve tasks that are beyond the reach of any single LLM. However, this potential can only be realized when the collaboration mechanism between agents is optimized. Specifically, optimizing the communication structure between agents is critical for fruitful collaboration. Most existing approaches rely on fixed topologies, pretrained graph generators, optimization over edges, or employ external LLM judges, thereby adding to the complexity. In this work, we introduce a *response-conditioned framework that adapts communication on-the-fly*. Agents independently generate responses to the user query and assess peer contributions using an approximation of the Shapley value. A directed acyclic graph (DAG) is then constructed to regulate the propagation of the responses among agents, which ensures stable and efficient message transmission from high-contributing agents to others. This graph is dynamically updated based on the agent responses from the previous collaboration round. Since *the proposed framework enables the self-organization of agents without additional supervision or training*, we refer to it as SELFORG. The SELFORG framework goes beyond task- and query-level optimization and takes into account the stochastic nature of agent responses. Experiments with both strong and weak LLM backends demonstrate robust performance, with significant gains in the weak regime where prior methods collapse. We also theoretically show that multiple agents increase the chance of correctness and that the correct responses naturally dominate the information flow.

## 1 Introduction

Large Language Models (LLMs) (OpenAI, 2023; Dubey et al., 2024; Anthropic, 2025; Qwen et al., 2025) have rapidly advanced capabilities across planning, analysis, coding, and dialog, yet a **single** LLM still faces notable limitations: stochastic or unreliable generations, hallucinations, and difficulty with long-horizon, multi-step tasks. A natural response has been to move from a solitary model to a **multi-agent system** (MAS) of LLMs, where agents interact, critique, and refine one another's outputs (Li et al., 2023; Chen et al., 2024; Zhuge et al., 2024; Qian et al., 2024b; Ye et al., 2025a). In principle, this collective can surpass an individual model by pooling complementary reasoning paths; in practice, however, the gains depend critically on **how** the agents are orchestrated: who communicates with whom, when, and how final outputs are aggregated.

Prior work has explored a spectrum of communication topologies. Fixed structures include chains, trees, complete graphs, and random graphs; scalable studies compare these patterns across task families such as mathematical reasoning, knowledge reasoning, and coding (Qian et al., 2025). Beyond static designs, some approaches treat the topology as **optimizable**: edges are sampled and trained with policy gradients or masks (e.g., GPTSwarm (Zhuge et al., 2024), AgentPrune (Zhang et al., 2025a)). A complementary line delegates topology design to a **separate** model that outputs a task/query-specific communication graph (e.g., G-Designer (Zhang et al., 2025b), MAS-GPT (Ye et al., 2025b). Others rely on an external LLM "judge" to rank, filter, or make final decisions (Ebrahimi et al., 2025). While effective in certain settings, these strategies introduce substantial overhead: pretraining a graph generator; reinforcement learning over edges; repeated calls to a judge LLM.

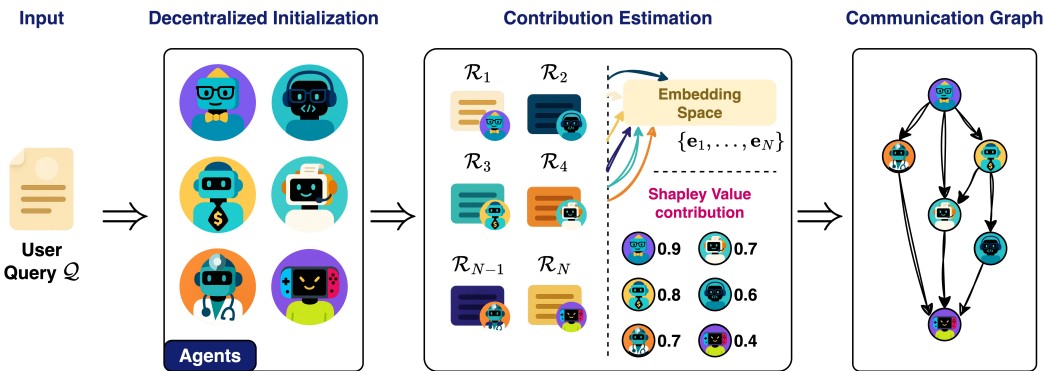

Figure 1: **Overview of SELFORG.** A query $\mathcal{Q}$ is distributed to $N$ agents, each producing a response $\mathcal{R}_n$. Responses are embedded, contributions estimated via Shapley-based valuation, and a directed acyclic communication graph is formed where edges reflect contributions and high-contribution agents lead. The figure depicts a single round; the process is iterated for $T$ rounds.

A common hypothesis in this literature is that there exists a "best" topology per **task category** (e.g., math vs. coding). This idea has evolved toward finer granularity, that the **query** should determine the topology (one graph per problem). We argue that both views are ultimately brittle. Because LLM agents are inherently stochastic, the information that matters for coordination is not the static task label nor the problem identity, but the **state** the agents are actually in – their concrete responses at a given time step. Two agents may answer the same query differently across runs; a topology that was ideal yesterday may be suboptimal today. Thus, the communication pattern should be decided **on the fly**, conditioned on the current pool of responses. Searching for a universally superior topology per task or per query is therefore potentially confounded and fragile: it risks overfitting to incidental response patterns or to powerful base models whose single-shot accuracy already masks orchestration weaknesses.

This state-driven perspective is especially revealing in the weak-backend regime, where each agent has a modest chance of being correct. In such settings, the value of orchestration should be to **amplify rare correct responses and suppress noise**, not to lean on an already-competent model. Our approach embraces this principle: we propose a decentralized, response-conditioned framework in which agents (i) independently produce initial answers, (ii) locally assess peers via a Shapley value-inspired contribution valuation, and (iii) construct a directed acyclic communication graph (DAG) that routes information from high-contribution agents to others. This yields a lightweight system with no external judge, no pretrained topology generator, and no edge-level reinforcement learning, yet it adapts its structure per instance.

We make the following contributions:

1. We construct a per-instance DAG directly from agents' current responses via semantic alignment, avoiding fixed topologies, pretrained graph generators, and edge-level RL.

2. We quantify influence with a Shapley-inspired utility, together with efficient approximation and ranking-stability guarantees, enabling lightweight, model-agnostic credit assignment.

3. We analyze why multi-agent interaction amplifies correct signals and why correct responders dominate contributions, and we validate SELFORG across various reasoning benchmarks and multiple backbones.

## 2 METHODOLOGY

We propose a multi-agent collaborative framework that adaptively constructs its communication structure without relying on external judges, pretrained graph generators, or reinforcement learning for edge optimization. The key principle is to leverage agents' own responses to estimate their contributions, estimate these contributions using Shapley values, and enforce a directed acyclic communication graph (DAG) for stable information propagation. In what follows, we describe each component in detail. The overall pipeline of SELFORG is illustrated in Figure 1.

## 2.1 System Overview

We formalize the collaboration in a multi-agent system as a dynamic directed graph $\mathcal{G}^{(t)} = (\mathcal{V}, \mathcal{E}^{(t)})$, where $\mathcal{V} = \{v_1, \ldots, v_N\}$ represents the set of nodes (with $|\mathcal{V}| = N$) and $\mathcal{E}^{(t)}$ denotes the set of edges in collaboration round $t \in [T]$. Each node $v_n \in \mathcal{V}$ represents an agent $\mathcal{A}_n$, instantiated with a backend LLM. Each agent $\mathcal{A}_n$ receives a prompt $\mathcal{P}_n^{(t)}$ and generates a response $\mathcal{R}_n^{(t)}$:

$$\mathcal{R}_n^{(t)} = \mathcal{A}_n(\mathcal{P}_n^{(t)}) = \mathcal{A}_n(\mathcal{P}_{n,\text{sys}}^{(t)}, \mathcal{P}_{n,\text{user}}, \mathcal{P}_{n,\text{coll}}^{(t)}), \tag{1}$$

where $\mathcal{P}_{n,\text{sys}}$ represents the system prompt that describes the agent's role and current state, $\mathcal{P}_{n,\text{user}}$ denotes the user prompt, which includes the given tasks, and $\mathcal{P}_{n,\text{coll}}$ includes responses from other agents (if available) and externally retrieved knowledge.

A directed edge $e_{m \to n}^{(t)} \in \mathcal{E}^{(t)}$ indicates that agent $\mathcal{A}_n$ incorporates information from agent $\mathcal{A}_m$ in round $t$. The presence (or absence) of an edge reflects the usefulness of $\mathcal{A}_m$'s response for $\mathcal{A}_n$. Thus, edges encode the information flow among agents. The graph can be equivalently expressed as an adjacency matrix $\mathbf{A}^{(t)} \in \{0,1\}^{N \times N}$, where $\mathbf{A}_{n,m}^{(t)} = 1$ if $e_{m \to n}^{(t)} \in \mathcal{E}^{(t)}$, otherwise 0.

## 2.2 Decentralized Initialization

This first stage of SELFORG (referred to as collaboration round $t = 0$) aims to generate a pool of diverse, but potentially noisy responses from $N$ agents. Given the user query $\mathcal{Q}$, each agent independently generates its own initial response $\mathcal{R}_n^{(0)}$. For this initial round, $\mathcal{P}_{n,\text{coll}}^{(0)} = \emptyset$ because agent $\mathcal{A}_n$ receives no input from other agents. We map each agent response $\mathcal{R}_n^{(0)}$ to an embedding $\mathbf{r}_n^{(0)} = f(\mathcal{R}_n^{(0)})$ with a lightweight model $f$ (e.g., `all-MiniLM-L6` (Reimers & Gurevych, 2019)), which need not be the same LLM used by the agents. These embeddings provide a fixed-dimensional, semantically meaningful representation of the agent responses. Subsequent stages use these response embeddings to infer contributions and construct the communication graph.

## 2.3 Contribution Estimation

Given responses $\{\mathbf{r}_1, \ldots, \mathbf{r}_N\}$ from the $N$ agents, we wish to estimate the contribution of individual agents towards generating the collective response. We frame the problem of contribution estimation as computing Shapley values (Shapley, 1953), a well-known concept in cooperative game theory. For a cooperative game, the Shapley value of agent $n$ is

$$\phi_n = \sum_{\mathcal{S} \subseteq [N] \setminus \{n\}} \frac{|\mathcal{S}|!(N - |\mathcal{S}| - 1)!}{N!} \left[ v(\mathcal{S} \cup \{n\}) - v(\mathcal{S}) \right]. \tag{2}$$

Here, $v(\mathcal{S})$ is the utility of coalition $\mathcal{S}$. Computing the true Shapley value using Eq. 2 requires $2^N$ evaluations, which is intractable for large $N$. Furthermore, an efficient mechanism is required to evaluate $v(\mathcal{S})$. This challenge is well-known in collaborative learning scenarios, where quantifying each player's contribution is crucial for tasks such as incentive mechanisms, fairness, and robustness (Lyu et al., 2020; Wang et al., 2020; Xu et al., 2021; Tastan et al., 2024; 2025a;b).

In this work, we adopt an approximation strategy inspired by Xu et al. (2021). Firstly, we define the utility of a coalition $\mathcal{S}$ as the cosine similarity between the average response embedding of the agents in $\mathcal{S}$ and the average response embedding of all agents. Moreover, instead of enumerating all coalitions, we compare each agent's embedding $\mathbf{r}_n$ directly against the average embedding $\mathbf{r}_{\text{avg}} = (1/N) \sum_{n=1}^N \mathbf{r}_n$. In other words, the true Shapley value $\phi_n$ is approximated by the estimated contribution $\psi_n$ of agent $\mathcal{A}_n$, which is defined as

$$\phi_n \approx \psi_n := \cos(\mathbf{r}_n, \mathbf{r}_{\text{avg}}). \tag{3}$$

The above approximation reduces the complexity of Shapley value computation from exponential to linear in $N$. Intuitively, the contribution is estimated based on how well an agent's response aligns with the collective (average) response. We now formalize the quality of this approximation.

**Theorem 1** (Approximation Bound (Xu et al., 2021))**.** *Suppose* $\|\mathbf{r}_n\| = \Gamma$ *for all* $n \in [N]$ *and* $|\langle \mathbf{r}_n, \mathbf{r}_{avg} \rangle| \geq 1/I$ *for some* $I > 0$. *Then*

$$\phi_n - L_n \psi_n \leq I\Gamma^2, \tag{4}$$

*where* $L_n$ *is a multiplicative factor that can be normalized away (Xu et al., 2021).*

**Corollary 1** (Ranking Stability)**.** *Let* $L_n$ *be the multiplicative factor from Theorem 1, and let* $\underline{L} = \min_j L_j$. *If*

$$\psi_n - \psi_k \; > \; \frac{2\,I\,\Gamma^2}{\underline{L}}, \tag{5}$$

*then the normalized Shapley scores* $\widetilde{\phi}_n = \phi_n/L_n$ *satisfy* $\widetilde{\phi}_n > \widetilde{\phi}_k$.

All proofs are deferred to the appendix. Thus, the approximate Shapley value $\psi_n$ not only provides an efficient approximation but also preserves the relative ordering of contributions when the separation between agents is sufficiently large.

## 2.4 COMMUNICATION GRAPH FORMATION

Given the current responses $\{\mathbf{r}_1^{(t)}, \ldots, \mathbf{r}_N^{(t)}\}$ from $N$ agents, our goal is to form a directed acyclic communication graph $\mathcal{G}^{(t+1)} = (\mathcal{V}, \mathcal{E}^{(t+1)})$ that governs how information flows among agents in the next round of collaboration $(t + 1)$. To form this graph, we first estimate the agent contributions as: $\psi_n^{(t+1)} = \cos(\mathbf{r}_n^{(t)}, \mathbf{r}_{\text{avg}}^{(t)})$. We also compute pairwise similarities between the agent responses by computing the cosine similarity between their response embeddings, i.e., $\mathbf{S}_{n,m}^{(t)} = \cos(\mathbf{r}_n^{(t)}, \mathbf{r}_m^{(t)})$.

To avoid a fully connected graph, we retain only semantically meaningful links: for agent $\mathcal{A}_n$, an incoming edge $e_{m \to n}^{(t+1)} \in \mathcal{E}^{(t+1)}$ is activated if and only if $\mathbf{S}_{n,m} \geq \tau$, where $\tau$ is a similarity threshold and $\psi_m^{(t+1)} > \psi_n^{(t+1)}$. Alternatively, sparse connectivity may be achieved by limiting active edges to the $k$ most similar neighbors of each agent.

---

**Algorithm 1** SELFORG

**Require:** Query $\mathcal{Q}$, similarity threshold $\tau$, optional neighbor budget $k$, total rounds $T$
**Ensure:** Final response $\mathcal{R}^\star$
1: $\mathcal{R}_n^{(0)} \leftarrow \mathcal{A}_n(\mathcal{Q}), \forall n \in [N]$
2: $(\mathcal{G}^{(0)}, \pi^{(0)}, \{\psi_n^{(0)}\}) \leftarrow$ ALG. 2($\{\mathcal{R}_n^{(0)}\}, \tau, k$)
3: **for** $t = 1$ to $T$ **do**
4:     **for** $n$ in $\pi^{(t-1)}$ **do**
5:         Collect $\{\mathcal{R}_m^{(t-1)} : e_{m \to n} \in \mathcal{E}^{(t-1)}\}$
6:         Form prompt $\mathcal{P}_n^{(t)} \leftarrow (\mathcal{Q}, \text{peer outputs})$
7:         Update response $\mathcal{R}_n^{(t)} \leftarrow \mathcal{A}_n(\mathcal{P}_n^{(t)})$
8:     **end for**
9:     $(\mathcal{G}^{(t)}, \pi^{(t)}, \{\psi_n^{(t)}\}) \leftarrow$ ALG. 2($\{\mathcal{R}_n^{(t)}\}, \tau, k$)
10:    Aggregate responses (Eq. 6)
11:    $\mathcal{R}^\star \leftarrow \arg\max_n \cos(\mathbf{r}_n^{(t)}, \mathbf{r}_{\text{centroid}}^{(t)})$
12:    Set $\mathcal{R}^\star$ as round output (fed to the leading agent in next round)
13: **end for**
14: **return** $\mathcal{R}^\star$ from final round $T$

---

The communication graph formed based on the above heuristics may still contain cycles. To avoid such cycles, we find the agent with the least estimated contribution within the detected cycle and remove the edge directed from the weaker agent (lower $\psi^{(t+1)}$) towards the stronger agent (higher $\psi^{(t+1)}$). This approach guarantees that more contributive agents remain upstream in the information flow. After the removal of the cycle, a topological ordering of the graph is computed, with ties broken in favor of nodes (agents) with higher $\psi^{(t+1)}$.

The resulting graph balances two principles:

    (i) *local alignment*, since each agent selectively listens only to semantically aligned peers, and

    (ii) *global reliability*, since contribution scores govern the final order and ensure correctness amplification.

Since most decisions regarding graph formation (except cycle detection and removal) are made locally, the resulting graph $\mathcal{G}$ is quite dynamic. Crucially, it is not predetermined by human design, but emerges from the content of the agent responses, embodying a form of *self-organizing team structure*. Each agent effectively votes on who should influence it, and the collective result is a network that channels information from the most promising agents to the ones that need help. For example,

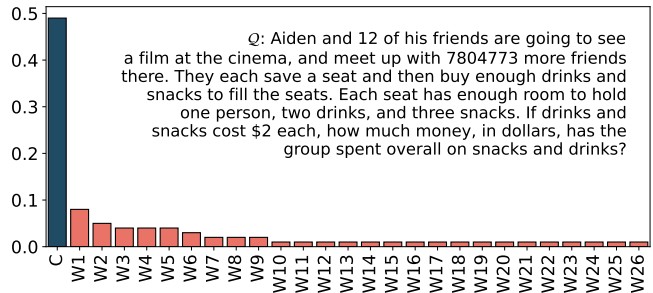 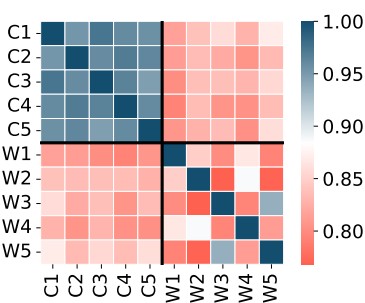

(a) Countplot of generated answers: the correct answer appears repeatedly, while wrong answers scatter across many alternatives with little agreement. Y-axis denotes no. of times the answer occurred.

(b) Heatmap plot of cosine similarities of embeddings from 5 correct and 5 wrong responses.

Figure 2: Analysis of Qwen-1.5B over 100 runs on the same math problem (GSM-Hard).

if one agent produces a particularly strong response and others recognize its value, many edges will point from the stronger agent to others, making it a hub of influence akin to a spontaneously elected leader. Thus, the topology adapts on-the-fly to the query at hand and the stochastic responses of the agents, rather than being fixed in advance. The full procedure is summarized in Algorithm 1.

## 2.5 RESPONSE PROPAGATION AND AGGREGATION

Once the communication graph $\mathcal{G}^{(t+1)}$ is formed, the next round of collaboration $(t+1)$ is initiated. There could be cases when the leader (root node) receives a message from the previous round (Algorithm 1, line 12) or it could coincide with its own response; in the latter case it is allowed to self-reflect on its previous response, i.e., $\mathcal{P}_{root,coll}^{(t+1)} \supseteq \mathcal{R}_{root}^{(t)}$. This ensures that the round begins with the most reliable response so far, while still leaving room for refinement. For the subsequent nodes in the graph, the response from the previous node is included in their collective prompt $\mathcal{P}_{n,coll}^{(t+1)} \supseteq \mathcal{R}_m^{(t+1)}$, if $e_{m \to n}^{(t+1)} = 1$. This response propagation procedure continues until all nodes in the current communication graph are processed. At the end of the response propagation, the agent contributions are re-estimated and the communication graph for the next collaboration round is formed. This process is repeated for a fixed number of collaboration rounds $T$ or until some early stopping criterion is met.

Thus, a multi-round procedure naturally emerges: (i) the first round establishes contributions and the influence structure, (ii) the highest-contributor's response initializes the next round, and (iii) subsequent agents refine or align their responses through the updated communication graph. In practice, two rounds are typically sufficient: the first for exploration, the second for consolidation.

After response propagation over multiple collaboration rounds, the final aggregate response of the multi-agent system is obtained as follows. First, the *contribution-weighted centroid* of the response embeddings after round $T$ is computed as:

$$\mathbf{r}_{centroid}^{(T)} = \frac{\sum_{n=1}^{N} \psi_n^{(T)} \mathbf{r}_n^{(T)}}{\sum_{n=1}^{N} \psi_n^{(T)}}, \tag{6}$$

where $\mathbf{r}_n^{(T)}$ is the response embedding of agent $\mathcal{A}_n$ in the last round and $\psi_n^{(T)}$ is its contribution score. The final aggregate response is not generated anew, but chosen among the existing responses $\{\mathcal{R}_n^{(T)}\}_{n=1}^N$. Specifically, we select the response whose embedding aligns closest to the centroid:

$$\mathcal{R}_{final} = \mathcal{R}_{n_\star}, \quad \text{where } n_\star = \arg\max_{n \in [N]} \cos\left(\mathbf{r}_n^{(T)}, \mathbf{r}_{centroid}^{(T)}\right). \tag{7}$$

## 2.6 PROBABILISTIC MODELING OF MULTI-AGENT SYSTEM

We now provide a probabilistic perspective to explain why our framework amplifies correct responses, particularly when the underlying LLMs are weak. The following analysis highlights two

complementary mechanisms: (i) with multiple agents, the probability that at least two agents are correct grows rapidly with $N$; and (ii) whenever multiple agents agree on the same response, that response is overwhelmingly likely to be correct. Together, these principles explain why correctness not only appears more often in multi-agent settings but also dominates the contribution scores.

We begin with the experiments in Figure 2. Figure 2a shows that while the correct answer consistently appears across 100 runs of Qwen-1.5B, wrong answers are scattered with little overlap. Panel 2b shows a cosine similarity of embeddings from 5 correct and 5 incorrect responses: correct answers form a tight cluster, whereas incorrect ones are scattered. Finally, an intervention study shows that when an agent receives input from the top-contributor, its probability of solving the task rises from 49% to 69%. These findings motivate the need for contribution estimation and leader selection in SELFORG.

If each agent independently answers correctly with probability $p \in (0, 1)$, then the probability that at least two of $N$ agents correct is $1 - (1 - p)^N - Np(1 - p)^{N-1}$. This is an increasing function with $N$ that quickly approaches 1. Therefore, even weak agents collectively increase the chance that agreement on correctness is present in the system. The role of SELFORG is then to identify these consensuses and amplify them. In the following straightforward lemma, we argue that consensus about a correct answer ($X_c$) is more likely than consensus about an incorrect answer ($X_i$) using observations from Figure 2.

**Lemma 1** (Agreement Concentration). *Let one agent be correct with probability $p \in (0, 1)$ and otherwise choose one of $K$ incorrect answers with probabilities $p_1, \ldots, p_K, \sum_{k=1}^{K} p_k = 1 - p$. For two independent agents,*

$$\Pr[X_c] = p^2 > \sum_{k=1}^{K} p_k^2 = \Pr[X_i]$$

*whenever the errors are sufficiently dispersed (as in Fig. 2a), e.g., $\max_k p_k \leq \frac{p^2}{1-p}$.*

We now connect the above probabilistic intuition to the contribution estimation of SELFORG. Figure 2b empirically supports the following assumption: embeddings of correct answers cluster together, while embeddings of wrong answers remain scattered.

**Assumption 1.** Suppose there exist constants $\alpha > \beta$ such that:

(i) For all $n, m \in \mathcal{S}$ (correct cluster), $\cos(\mathbf{r}_n, \mathbf{r}_m) \geq \alpha$;

(ii) For all $n \in \mathcal{S}, k \notin \mathcal{S}, \cos(\mathbf{r}_n, \mathbf{r}_k) \leq \beta$,

(iii) For all $k, \ell \notin \mathcal{S}, \cos(\mathbf{r}_k, \mathbf{r}_\ell) \leq \beta$,

**Lemma 2** (Contribution Dominance). *Under Assumption 1, for every $n \in \mathcal{S}$ and $k \notin \mathcal{S}$ we have $\psi_n > \psi_k$, where $\psi_n = \cos(\mathbf{r}_n, \mathbf{r}_{\text{all}})$ is the contribution score.*

Lemmas 1 and 2 together yield the following guarantee:

**Corollary 2** (Correctness Amplification). *If at least two agents output the correct response, then this response is strictly more likely to receive high contribution scores than any incorrect alternative. The communication graph, therefore, routes information preferentially from correct agents, amplifying their signals while suppressing noise.*

Together, these results formalize why SELFORG remains effective under the weak-backend regime.

## 3 EXPERIMENTS

Our empirical evaluation largely follows the MASLab benchmark protocol (Ye et al., 2025a). We test SELFORG across various LLM backbones: Qwen (Qwen-2.5-{1.5, 3, 7, 14, 32, 72}B) (Qwen et al., 2025), LLaMA (LLaMA-3-8B-Instruct, LLaMA-3.3-70B-Instruct) (Dubey et al., 2024), Falcon (Falcon3-7B-Instruct) (TII, 2024; Almazrouei et al., 2023), and Mistral (Mistral-7B-Instruct-v0.3) (Jiang et al., 2023a) on mathematics (MATH (Hendrycks et al., 2021), GSM8K (Cobbe et al., 2021), GSM-Hard (Gao et al., 2023), AQUA-RAT (Ling et al., 2017), AIME-2024), science (GPQA (Rein et al., 2024)), and knowledge (MMLU (Hendrycks et al., 2021), MMLU-Pro (Wang

Table 1: **Main results on Qwen-2.5-1.5B-Instruct.** Comparison of SELFORG with single-agent prompting and multi-agent baselines across seven reasoning benchmarks. AVG reports mean accuracy, while AVG-R reports average rank across methods (lower is better).

| Method | MATH | GSM8K | AQUA | GSM-H | MMLU | MMLU-P | AIME | AVG | AVG-R |
|---|---|---|---|---|---|---|---|---|---|
| Qwen-2.5-1.5B-Instruct | | | | | | | | | |
| Single | 49.20 | 70.40 | 51.18 | 36.20 | 49.60 | 28.80 | 3.33 | 41.24 | 2.57 |
| CoT | 46.80 | 69.20 | 53.54 | 36.20 | 50.60 | 28.60 | 3.33 | 41.18 | 2.71 |
| DyLAN | 49.80 | 67.80 | 51.18 | 27.20 | 50.00 | 15.40 | 3.33 | 37.82 | 4.00 |
| MacNet | 45.40 | 64.20 | 49.21 | 29.40 | 42.00 | 26.00 | 0.00 | 36.60 | 4.57 |
| G-Designer | 42.20 | 61.40 | 44.48 | 24.20 | 40.00 | 22.00 | 0.00 | 33.47 | 5.86 |
| AgentVerse | 45.20 | 69.00 | 50.39 | 27.80 | 38.20 | 24.00 | 0.00 | 36.37 | 4.86 |
| AutoGen | 11.60 | 69.40 | 28.74 | 5.40 | 12.20 | 5.20 | 0.00 | 18.93 | 6.06 |
| SELFORG | **52.40** | **74.60** | **58.27** | **38.00** | **53.80** | **31.60** | **6.67** | **45.05** | **1.00** |

Table 2: **Main results on large models (LLaMA-3.3-70B-Instruct & Qwen-2.5-72B-Instruct).** Comparison of SELFORG with baselines across reasoning benchmarks. AVG reports mean accuracy and AVG-R reports average rank across methods (lower is better).

| Method | MATH | GSM8K | AQUA | GSM-H | MMLU | MMLU-P | GPQA | AIME | AVG | AVG-R |
|---|---|---|---|---|---|---|---|---|---|---|
| LLaMA-3.3-70B-Instruct | | | | | | | | | | |
| Single | 74.80 | 96.20 | 77.56 | 54.00 | 84.40 | 68.40 | 55.36 | 23.33 | 66.76 | 3.88 |
| CoT | 75.00 | 95.80 | 79.92 | **57.40** | **85.20** | 71.00 | 56.70 | 26.67 | 68.46 | 2.50 |
| DyLAN | 77.60 | 95.20 | 76.38 | 53.00 | 83.60 | 31.60 | 58.04 | 26.67 | 62.76 | 4.25 |
| MacNet | 74.80 | 96.00 | 79.13 | 55.20 | 83.00 | 65.40 | 58.26 | 26.67 | 67.31 | 3.63 |
| AgentVerse | 76.80 | 94.60 | 76.38 | 51.20 | 83.60 | 69.20 | 55.36 | 26.67 | 66.73 | 4.50 |
| AutoGen | 70.80 | 93.00 | 79.50 | 51.40 | 82.60 | 64.60 | 52.68 | **30.00** | 65.57 | 5.13 |
| SELFORG | **79.80** | **96.60** | **81.10** | 56.80 | 85.00 | **72.40** | 59.82 | 30.00 | **70.19** | **1.25** |
| Qwen-2.5-72B-Instruct | | | | | | | | | | |
| Single | 83.00 | 95.00 | **81.10** | 63.80 | 82.40 | 70.60 | 46.65 | 20.00 | 67.82 | 2.88 |
| CoT | 82.80 | 95.20 | 80.71 | 62.00 | 82.80 | **71.40** | 44.20 | 16.67 | 66.97 | 3.50 |
| DyLAN | 80.60 | 95.40 | 77.95 | 63.20 | **84.20** | 69.20 | 46.43 | 13.33 | 66.29 | 3.75 |
| MacNet | 81.40 | 95.40 | 79.13 | 62.80 | 83.20 | 65.60 | 40.40 | 16.67 | 65.58 | 4.13 |
| AgentVerse | 82.80 | 95.20 | 77.17 | 57.80 | 81.40 | 71.20 | 45.98 | **23.33** | 66.86 | 4.13 |
| AutoGen | 81.20 | 95.80 | 78.35 | 64.20 | 82.60 | 69.40 | 45.54 | 13.33 | 66.30 | 3.75 |
| SELFORG | **84.40** | **96.20** | 80.71 | **64.20** | 83.80 | 71.20 | **47.77** | 23.33 | **68.95** | **1.38** |

et al., 2024)) benchmarks. We set the default max token limit as 2048 and a temperature 0.5. Our baselines include single call, chain-of-thought (CoT) (Wei et al., 2022), AutoGen (Wu et al., 2024), AgentVerse (Chen et al., 2024), G-Designer (Zhang et al., 2025b), DyLAN (Liu et al., 2024), and MacNet (Qian et al., 2025). SELFORG defaults to use $N = 4$ agents, top-2 neighbors and at most 3 rounds. Additional configurations, baseline methods, and other details are provided in Appendix B.

## 3.1 MAIN EXPERIMENTAL RESULTS

Table 1 highlights the key advantage of SELFORG in scenarios where orchestration is most challenging. With Qwen-1.5B, all multi-agent baselines cluster around average accuracies of roughly $33 - 37\%$, showing limited ability to harness collaboration when the underlying agents are weak. In contrast, SELFORG achieves an average accuracy of $45.05\%$, a clear margin above all baselines, while also attaining the best average rank (AVG-R). This represents a gain of nearly $+4$ **points** over the strongest non-collaborative baseline (single agent or CoT). These results confirm our central hypothesis: when responses are noisy and correctness is sparse, a response-conditioned, adaptive graph provides the necessary amplification mechanism to elevate correct signals and suppress noise. We include G-Designer at a small scale; see Appendix B for discussion.

We also test SELFORG on stronger backbone models (Table 2). For LLaMA-70B, SELFORG achieves the highest average accuracy ($70.19\%$) and best AVG-R ($1.25$), outperforming all baselines. The same holds for the Qwen-72B model, where SELFORG attains the best average rank ($1.38$) with clear gains over prior methods. These results demonstrate that SELFORG remains effective even with frontier-scale models, providing complementary improvements.

| Dataset | AQUA-RAT | | MMLU-Pro | |
|---|---|---|---|---|
| Model | Single | SELFORG | Single | SELFORG |
| 1.5B | 51.18 | 58.27 | 28.80 | 31.60 |
| 3B | 65.35 | 73.62 | 42.60 | 46.20 |
| 7B | 73.62 | 78.35 | 53.20 | 56.40 |
| 14B | 75.79 | 81.50 | 61.80 | 65.40 |
| 32B | 79.53 | 83.07 | 67.40 | 70.20 |
| 72B | 81.10 | 80.71 | 70.60 | 71.20 |

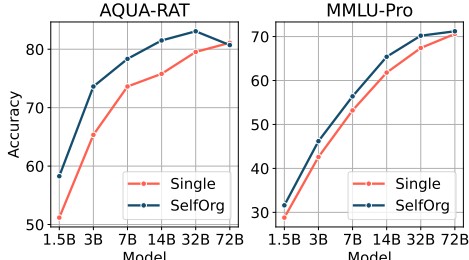

Figure 3: Scaling laws of **Qwen-2.5-X-Instruct** models across two reasoning benchmarks (AQUA-RAT and MMLU-Pro). The table shows exact accuracy values for different model sizes under the Single and SELFORG settings, while plot visualizes performance trends.

Together, these results demonstrate that SELFORG consistently outperforms prior orchestration frameworks. Gains are most pronounced in the low-capacity regime, where amplification of correct signals is crucial, but remain competitive even for frontier-scale models.

## 3.2 SCALING LAWS

We analyze how SELFORG scales with model size by evaluating Qwen-2.5-X-Instruct models ranging from 1.5B to 72B parameters on AQUA-RAT and MMLU-Pro (Table 3). Across most sizes, SELFORG consistently improves over the single-agent baseline. For example, gains are most pronounced in the weak-to-medium regime, with the 3B model improving from 65.35 to 73.62 on AQUA-RAT and from 42.60 to 46.20 on MMLU-Pro. At larger scales, improvements persist but become smaller, reflecting that strong single agents already achieve high reliability.

Interestingly, at the extreme high end (72B), the benefit nearly vanishes on AQUA-RAT, where accuracy slightly decreases from 81.10 to 80.71. This suggests diminishing returns when base models are sufficiently strong that agreement across agents offers limited additional signal. Nevertheless, SELFORG never underperforms substantially, and its advantages are clearest when individual models are weak or moderately strong, confirming the theoretical expectation that multi-agent collaboration amplifies correctness most in the low-resource regime.

## 3.3 HETEROGENEOUS AGENTS

We evaluate SELFORG in settings where agents are instantiated with heterogeneous backbones: Qwen2.5-7B, Falcon3-7B, Llama-3-8B, and Mistral-7B. Although similar in parameter count, these models differ substantially in ability (Table 4, top), with Qwen strongest, Mistral weakest, and Falcon serving as the second-best. Since multi-agent success depends on agreement among strong contributors, the system's performance is effectively bounded by Falcon's reliability while aiming to approach Qwen's level.

| Model | AQUA-RAT | MMLU-Pro |
|---|---|---|
| Qwen | 76.38 | 51.60 |
| Falcon | 61.42 | 47.00 |
| LLaMA | 44.09 | 40.60 |
| Mistral | 25.20 | 26.80 |

| **Single** (↑) / **AQUA-RAT** (↓) | | |
|---|---|---|
| Model | Single | SELFORG |
| Mix | 53.94 | 66.14 |
| Mix | 41.60 | 50.40 |

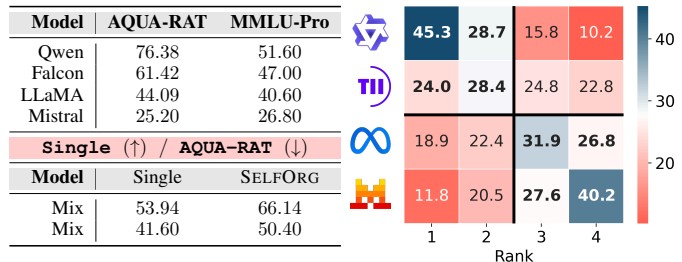

Figure 4: **Heterogeneous Agents.** Left: accuracies on AQUA-RAT and MMLU-Pro for each backbone and for the mixed-pool baseline (Single) vs. SELFOR. Right: percentage of times each agent attains contribution rank $r$ (rank-1 highest).

The lower part of Table 4 compares the Single baseline (where one model is randomly sampled per query) and SELFORG. The Single setting yields 53.94 accuracy on AQUA-RAT and 41.60 on MMLU-Pro, whereas SELFORG improves to 66.14 and 50.40. Thus, SELFORG leverages agreement between strong models while still extracting useful signals from weaker ones, outperforming the stochastic baseline and approaching the best single-agent.

Contribution rank distributions (Figure 4) further illustrate this effect: Qwen and Falcon dominate higher ranks, while LLaMA and Mistral are usually relegated lower, though occasionally contributing at mid-rank when aligned with stronger peers.

We further evaluate configurations that mix strong and weak agents, with detailed results presented in Appendix D.1. Beyond accuracy, we also analyze efficiency in terms of token usage (Appendix D.2). Additional ablation studies examine the impact of the number of agents, the effect of reform across rounds, and the role of the embedding model in contribution estimation (Appendix F).

## 4 Related Work

**Multi-Agent Systems.** Early multi-agent systems such as CAMEL (Li et al., 2023) and AutoGen (Wu et al., 2024) introduced role-based LLM agents that collaborate through dialogue. Debate-style systems encourage adversarial or diverse reasoning to refine answers (Du et al., 2023; Liang et al., 2024; Subramaniam et al., 2025), while dynamic orchestration (AgentVerse (Chen et al., 2024), DyLAN (Liu et al., 2024)) adapts team composition or roles during execution. More recent efforts aim for automatic workflow generation (Hu et al., 2025; Zhang et al., 2025c;b; Ye et al., 2025b), though these rely on strong meta-agents or pretrained generators, adding overhead and limiting autonomy. Multi-agent collaboration has also been applied to diverse domains including software (Hong et al., 2024; Qian et al., 2024a), recommendation (Zhang et al., 2024), medicine (Tang et al., 2024), finance (Li et al., 2024), education (Zhang et al., 2025e), and science (Zeng et al., 2024).

**Communication Graphs.** Prior work has explored a spectrum of communication topologies. Fixed structures include chains, trees, complete graphs, and random graphs, with recent studies systematically comparing these patterns across task families such as mathematical reasoning, knowledge reasoning, and coding (Qian et al., 2025). Beyond static designs, some approaches treat the topology as *optimizable*: edges are sampled and trained with policy gradients or masks (Zhuge et al., 2024; Zhang et al., 2025a; Qian et al., 2025). A complementary line delegates topology design to a *separate* model that outputs a task- or query-specific communication graph (Zhang et al., 2025b; Ye et al., 2025b). Other frameworks rely on an external LLM "judge" to rank, filter, or finalize outputs (Liu et al., 2024; Zhang et al., 2025c; Zhuge et al., 2025; Ebrahimi et al., 2025), or extend to decentralized settings (Yang et al., 2025; Lu et al., 2024). While effective in constrained settings, these strategies incur substantial overhead: pretraining graph generators, optimization over edges, or repeated calls to a judge LLM.

These approaches assume that an optimal or near-optimal graph exists either per task category or even per query. However, such assumptions can be misleading: because LLM agents are stochastic, the same agent may succeed on one query and fail on another. Our method instead constructs the graph on-the-fly, adapting dynamically to the actual responses produced.

**Contribution Assessment in Collaborative Systems.** Numerous systems in LLM-based MAS assess agent quality with additional LLMs. For instance, LLM-Blender (Jiang et al., 2023b) uses an additional LLM for pairwise comparisons, incurring $\mathcal{O}(N^2)$ operations for $N$ agents, while DyLAN (Liu et al., 2024) introduces a dedicated LLM agent to score responses; other MAS frameworks similarly rely on judge models to value and select contributions (Ebrahimi et al., 2025). Outside multi-agent systems, the broader literature on contribution valuation offers principled tools originating from cooperative game theory (Shapley, 1953), with concrete instantiations in federated learning (McMahan et al., 2017; Jia et al., 2019). FL works measure participant contributions via Shapley values (Jia et al., 2019; Xu et al., 2021; Liu et al., 2022; Tastan et al., 2024), influence functions (Rokvic et al., 2024), self-reported information (Kang et al., 2019), and utility-game formulations (Wang et al., 2019). We draw a direct parallel to MAS and instantiate Shapley-style contribution estimates over agent responses (Section 2.3), eliminating external judges and additional training while maintaining principled contribution estimation.

## 5 Conclusion

We presented SELFORG, a framework for orchestrating LLM-based multi-agent systems without external pretrained topology generators or reinforcement learning. By leveraging response-conditioned contribution estimation and adaptive graph formation, SELFORG amplifies correct signals and suppresses noise. Our theoretical analysis and empirical results across diverse reasoning benchmarks confirm that it consistently outperforms prior orchestration baselines.

ACKNOWLEDGMENTS

This material is partly based on work supported by the Office of Naval Research N00014-24-1-2168.

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

# Contents

# A MATHEMATICAL PROOFS

## A.1 PROOF OF THEOREM 1

*Proof.* We adapt the argument of (Xu et al., 2021) to our setting.

By definition,

$$\phi_n = \sum_{\mathcal{S} \subseteq [N] \setminus \{n\}} w_{\mathcal{S}} \Delta_n(\mathcal{S}), \quad \Delta_n(\mathcal{S}) = v(\mathcal{S} \cup \{n\}) - v(\mathcal{S}), \quad w_{\mathcal{S}} = \frac{|\mathcal{S}|!(N - |\mathcal{S}| - 1)!}{N!}. \quad (8)$$

Let $\boldsymbol{x} = \sum_{j \in \mathcal{S}} \mathbf{r}_m$ and recall $\mathbf{r}_{\text{avg}} = \frac{1}{N} \sum_{j=1}^{N} \mathbf{r}_m$.

**Exact decomposition.** Expanding the marginal contribution (difference in the utilities) $\Delta_n(\mathcal{S})$ and regrouping gives

$$\Delta_n(\mathcal{S}) \quad = \quad v(\mathcal{S} \cup \{n\}) - v(\mathcal{S}) \tag{9}$$

$$= \quad \frac{\langle \boldsymbol{x} + \mathbf{r}_n, \mathbf{r}_{\text{avg}} \rangle}{\|\boldsymbol{x} + \mathbf{r}_n\| \|\mathbf{r}_{\text{avg}}\|} - \frac{\langle \boldsymbol{x}, \mathbf{r}_{\text{avg}} \rangle}{\|\boldsymbol{x}\| \|\mathbf{r}_{\text{avg}}\|} \tag{10}$$

$$= \quad \frac{1}{\|\mathbf{r}_{\text{avg}}\|} \left( \frac{\langle \boldsymbol{x}, \mathbf{r}_{\text{avg}} \rangle}{\|\boldsymbol{x} + \mathbf{r}_n\|} - \frac{\langle \boldsymbol{x}, \mathbf{r}_{\text{avg}} \rangle}{\|\boldsymbol{x}\|} + \frac{\langle \mathbf{r}_n, \mathbf{r}_{\text{avg}} \rangle}{\|\boldsymbol{x} + \mathbf{r}_n\|} \right) \tag{11}$$

$$= \quad \frac{1}{\|\mathbf{r}_{\text{avg}}\|} \left( \frac{\|\boldsymbol{x}\| - \|\boldsymbol{x} + \mathbf{r}_n\|}{\|\boldsymbol{x} + \mathbf{r}_n\|} \cdot \frac{\langle \boldsymbol{x}, \mathbf{r}_{\text{avg}} \rangle}{\|\boldsymbol{x}\|} + \frac{\langle \mathbf{r}_n, \mathbf{r}_{\text{avg}} \rangle}{\|\boldsymbol{x} + \mathbf{r}_n\|} \right) \tag{12}$$

$$= \quad \frac{\|\boldsymbol{x}\| - \|\boldsymbol{x} + \mathbf{r}_n\|}{\|\boldsymbol{x} + \mathbf{r}_n\|} \frac{\langle \boldsymbol{x}, \mathbf{r}_{\text{avg}} \rangle}{\|\boldsymbol{x}\| \|\mathbf{r}_{\text{avg}}\|} + \frac{1}{\|\boldsymbol{x} + \mathbf{r}_n\|} \frac{\langle \mathbf{r}_n, \mathbf{r}_{\text{avg}} \rangle}{\|\mathbf{r}_{\text{avg}}\|} \tag{13}$$

$$= \quad \underbrace{\frac{\|\boldsymbol{x}\| - \|\boldsymbol{x} + \mathbf{r}_n\|}{\|\boldsymbol{x} + \mathbf{r}_n\|}}_{A_{\mathcal{S}}} \cdot v(\mathcal{S}) + \underbrace{\frac{\|\mathbf{r}_n\|}{\|\boldsymbol{x} + \mathbf{r}_n\|}}_{B_{\mathcal{S}}} \cdot \psi_n \tag{14}$$

where $v(\mathcal{S}) = \cos(\boldsymbol{x}, \mathbf{r}_{\text{avg}})$ and $\psi_n = \cos(\mathbf{r}_n, \mathbf{r}_{\text{avg}})$. $A_{\mathcal{S}} = \dfrac{\|\boldsymbol{x}\| - \|\boldsymbol{x} + \mathbf{r}_n\|}{\|\boldsymbol{x} + \mathbf{r}_n\|}$ and $B_{\mathcal{S}} = \dfrac{\|\mathbf{r}_n\|}{\|\boldsymbol{x} + \mathbf{r}_n\|}$.

Plugging this back into the original equation of Shapley value gives the exact split

$$\phi_n = \sum_{\mathcal{S}} w_{\mathcal{S}} A_{\mathcal{S}} v(\mathcal{S}) + \left[ \sum_{\mathcal{S}} w_{\mathcal{S}} B_{\mathcal{S}} \right] \psi_n = L_n \psi_n + \sum_{\mathcal{S}} w_{\mathcal{S}} A_{\mathcal{S}} v(\mathcal{S}). \tag{15}$$

**Bounding the error.** Consider the ratio

$$\frac{|A_{\mathcal{S}}| |v(\mathcal{S})|}{B_{\mathcal{S}} \psi_n} = \frac{|\|\boldsymbol{x}\| - \|\boldsymbol{x} + \mathbf{r}_n\||}{\Gamma} \cdot \frac{|\cos(\boldsymbol{x}, \mathbf{r}_{\text{avg}})|}{\cos(\mathbf{r}_n, \mathbf{r}_{\text{avg}})}. \tag{16}$$

Using (i) the reverse triangle inequality $|\|\boldsymbol{x}\| - \|\boldsymbol{x} + \mathbf{r}_n\|| \leq \|\mathbf{r}_n\| = \Gamma$, (ii) $|\cos(\boldsymbol{x}, \mathbf{r}_{\text{avg}})| \leq 1$, and (iii) the alignment assumption ($|\langle \mathbf{r}_n, \mathbf{r}_{\text{avg}} \rangle| \geq \frac{1}{I}$), we obtain

$$\frac{|A_{\mathcal{S}}| |v(\mathcal{S})|}{B_{\mathcal{S}} \psi_n} \leq I \Gamma \|\mathbf{r}_{\text{avg}}\| \leq I \Gamma^2, \tag{17}$$

using $\|\mathbf{r}_{\text{avg}}\| \leq \Gamma$ (average of $\Gamma$-norm vectors). Averaging with weights $w_{\mathcal{S}}$ (linear interpolation in our case) preserves this bound, yielding

$$\phi_n - L_n \psi_n \leq I \Gamma^2. \tag{18}$$

This concludes the proof.

$\square$

## A.2 PROOF OF COROLLARY 1

*Proof.* From Theorem 1, we can write

$$\widetilde{\phi}_\ell = \psi_\ell + \frac{R_\ell}{L_\ell}, \quad |R_\ell| \le I\Gamma^2. \tag{19}$$

Then,

$$\widetilde{\phi}_n - \widetilde{\phi}_k \ge (\psi_n - \psi_k) - \frac{|R_n|}{L_n} - \frac{|R_k|}{L_k} \ge (\psi_n - \psi_k) - \frac{2I\Gamma^2}{\underline{L}}. \tag{20}$$

Hence, if $\psi_n - \psi_k > 2I\Gamma^2/\underline{L}$, then $\widetilde{\phi}_n > \widetilde{\phi}_k$.

□

## A.3 PROOF OF LEMMA 1

*Proof.* By independence, $\Pr[X_\mathrm{c}] = p^2$ and $\Pr[X_\mathrm{i}] = \sum_k p_k^2$. Using dispersion,

$$\sum_{k=1}^{K} p_k^2 \le (\max_k p_k) \sum_{k=1}^{K} p_k = (1-p) \max_k p_k \le (1-p)\frac{p^2}{1-p} = p^2. \tag{21}$$

Strict inequality holds unless all mass concentrates on a single incorrect option at exactly $\max_k p_k = \frac{p^2}{1-p}$. Hence, agreement is more likely on the correct answer.

This completes the proof.

□

## A.4 PROOF OF LEMMA 2

*Proof.* Fix $n \in \mathcal{S}$. Decompose

$$\langle \mathbf{r}_n, \mathbf{r}_{\mathrm{avg}} \rangle = \langle \mathbf{r}_n, \mathbf{r}_n \rangle + \sum_{\substack{m \in \mathcal{S} \\ m \ne n}} \langle \mathbf{r}_n, \mathbf{r}_m \rangle + \sum_{u \notin \mathcal{S}} \langle \mathbf{r}_n, \mathbf{r}_u \rangle. \tag{22}$$

By assumptions (i)-(ii),

$$\langle \mathbf{r}_n, \mathbf{r}_n \rangle = \Gamma^2, \qquad \langle \mathbf{r}_n, \mathbf{r}_m \rangle \ge \Gamma^2 \alpha \ (m \in \mathcal{S} \setminus \{n\}), \qquad \langle \mathbf{r}_n, \mathbf{r}_u \rangle \le \Gamma^2 \beta \ (u \notin \mathcal{S}). \tag{23}$$

Hence

$$\langle \mathbf{r}_n, \mathbf{r}_{\mathrm{avg}} \rangle \ge \Gamma^2 + (|\mathcal{S}| - 1) \Gamma^2 \alpha + (N - |\mathcal{S}|) \Gamma^2 \beta. \tag{24}$$

Now fix $k \notin \mathcal{S}$. Similarly,

$$\langle \mathbf{r}_k, \mathbf{r}_{\mathrm{avg}} \rangle = \langle \mathbf{r}_k, \mathbf{r}_k \rangle + \sum_{v \in \mathcal{S}} \langle \mathbf{r}_k, \mathbf{r}_v \rangle + \sum_{\substack{w \notin \mathcal{S} \\ w \ne k}} \langle \mathbf{r}_k, \mathbf{r}_w \rangle. \tag{25}$$

By assumptions (ii)-(iii),

$$\langle \mathbf{r}_k, \mathbf{r}_{\mathrm{avg}} \rangle \le \Gamma^2 + |\mathcal{S}| \Gamma^2 \beta + (N - |\mathcal{S}| - 1) \Gamma^2 \beta = \Gamma^2 + (N - 1) \Gamma^2 \beta. \tag{26}$$

Subtracting yields

$$\langle \mathbf{r}_n, \mathbf{r}_{\mathrm{avg}} \rangle - \langle \mathbf{r}_k, \mathbf{r}_{\mathrm{avg}} \rangle \ge (|\mathcal{S}| - 1) (\alpha - \beta) \Gamma^2 > 0. \tag{27}$$

Since all $\psi_r = \cos(\mathbf{r}_r, \mathbf{r}_{\mathrm{avg}})$ share the same denominator $\|\mathbf{r}_r\| \|\mathbf{r}_{\mathrm{avg}}\| = \Gamma \|\mathbf{r}_{\mathrm{avg}}\|$, the inequality implies $\psi_n > \psi_k$.

This completes the proof.

□

## B   IMPLEMENTATION DETAILS

**Baselines.**   We use the benchmark authors' implementations where available (Ye et al., 2025a).

- **MacNet** (Qian et al., 2025) is run with 5 agents and the random topology, following the paper's strongest reported configuration.
- **DyLAN** (Liu et al., 2024) uses 4 agents and 3 rounds.
- **AgentVerse** (Chen et al., 2024) and **AutoGen** (Wu et al., 2024) are run with their public defaults adapted to the benchmark.
- **G-Designer** (Zhang et al., 2025b) is evaluated on Qwen-2.5-1.5B-Instruct; we omit larger models because it requires training a separate graph generator, and thus latency-inefficient (see Sections 1 and 4 for discussion).

    We include G-Designer (Zhang et al., 2025b) in our Qwen-1.5B experiments, as it is among the most closely related graph-optimizing methods. However, its design differs fundamentally from SELFORG. G-Designer trains a separate graph generator that outputs a communication topology conditioned on the query and predefined agent roles. While this is effective with stronger backbones, it does not adapt to the *responses* actually produced by weak agents, which are often noisy. As a result, its learned graphs fail to amplify correct signals in the low-capacity regime, leading to poor empirical performance (see Table 1).

    For larger models, we do not run G-Designer, since it requires training a dedicated graph generator. This introduces substantial overhead and deviates from our goal of efficient, judge-free orchestration. Our design philosophy emphasizes lightweight, response-conditioned self-organization without external generators or meta-agents, as discussed in Sections 1 and 4.

- To compare with **single agent execution methods**, we incorporate evaluations against single execution and chain-of-thought (CoT) prompting (Wei et al., 2022).

**SELFORG configuration.**   Implementation can be found at: https://github.com/tnurbek/selforg. SELFORG is configured as:

- **Agent pool:** {`Assistant`, `Programmer`, `Mathematician`, `Economist`, `Psychologist`, `Historian`, `Lawyer`, `Doctor`}.
- **Number of agents:** for math-based tasks: 4 agents with fixed roles (from the pool), and for science and knowledge: 5 agents up to psychologist.
- **Neighbor selection:** top-2 neighbors per agent (by pairwise cosine similarity $\mathbf{S}$); similarity threshold $\tau = 0.5$ for edge formation.
- **Rounds and structure:** a maximum of 3 rounds (including decentralized initialization); with DAG enforcement.
- **Contribution estimation:** we use `all-MiniLM-L6-v2` embedding model with an embedding dimension of $384$ (a lightweight sentence embedding model).
- **Aggregation:** contribution-weighted centroid (Equation 6); final answer is the nearest response to the centroid.
- **Reform policy:** we reform the DAG in each round based on the updated responses.

**Agent Profiling.**   We adopt a standard community template for defining agent roles that is widely used in prior multi-agent system benchmarks. In our experiments, a subset of four/five agents is instantiated per run (default), selected in fixed order unless otherwise specified. Each role is assigned a default prompt template (system instruction) from the benchmark community, without additional fine-tuning or hand-engineering. This ensures that performance differences arise from orchestration rather than custom role design.

The role descriptions are provided below.

**Evaluation.**   We use a direct scoring approach with a task-specific evaluator (xVerify (Chen et al., 2025)), which is fine-tuned to assess correctness across various domains (Ye et al., 2025a).

**Assistant**

You are a super-intelligent AI assistant capable of performing tasks more effectively than humans.

**Mathematician**

You are a mathematician.
You are good at math games, arithmetic calculation, and long-term planning.

**Economist**

You are an economist.
You are good at economics, finance, and business. You have experience on understanding charts while interpreting the macroeconomic environment prevailing across world economies.

**Psychologist**

You are a psychologist.
You are good at psychology, sociology, and philosophy. You give people scientific suggestions that will make them feel better.

**Programmer**

You are a programmer.
You are good at computer science, engineering, and physics. You have experience in designing and developing computer software and hardware.

**Historian**

You are a historian.
You research and analyze cultural, economic, political, and social events in the past, collect data from primary sources and use it to develop theories about what happened during various periods of history.

**Lawyer**

You are a lawyer.
You are good at law, politics, and history.

**Doctor**

You are a doctor and come up with creative treatments for illnesses or diseases. You are able to recommend conventional medicines, herbal remedies and other natural alternatives. You also consider the patient's age, lifestyle and medical history when providing your recommendations.

## C  GRAPH FORMATION FUNCTION

---

**Algorithm 2** Graph Formation

---

**Require:** Responses $\{\mathcal{R}_n\}_{n=1}^N$, similarity threshold $\tau$, optional neighbor budget $k$
**Ensure:** Graph $\mathcal{G} = (\mathcal{V}, \mathcal{E})$, topological order $\pi$, contribution scores $\{\psi_n\}_{n=1}^N$
 1: Compute embeddings $\mathbf{r}_n \leftarrow f(\mathcal{R}_n)$, $\forall n \in [N]$
 2: Form similarity matrix $\mathbf{S}$
 3: Get contribution scores $\{\psi_n\}_{n=1}^N$ (Eq. 3)
 4: Initialize edge set $\mathcal{E} \leftarrow \{\}$
 5: **for** $n = 1$ to $N$ **do**
 6:     $\mathcal{N} \leftarrow \{m \neq n : \mathbf{S}_{n,m} \geq \tau\}$
 7:     **if** $k$ specified **then**
 8:         keep top-$k$ in $\mathcal{N}$
 9:     **end if**
10:     **for** $m \in \mathcal{N}$ **do**
11:         add edge $e_{m \rightarrow n}$ to $\mathcal{E}$
12:     **end for**
13: **end for**
14: **while** $\mathcal{E}$ contains a cycle **do**
15:     Identify cycle $\mathcal{C}$
16:     Remove edge from lower-$\psi$ to higher-$\psi$ node in $\mathcal{C}$
17: **end while**
18: Obtain topological order $\pi$ of $\mathcal{G} = (\mathcal{V}, \mathcal{E})$
19: **return** $(\mathcal{G}, \pi, \{\psi_n\})$

---

## D  ADDITIONAL EXPERIMENTS

### D.1  WEAK AGENT IN A POOL

To test the robustness of SELFORG in a setting with a weak agent present, we evaluate configurations where weaker agents are introduced alongside stronger peers. Figure 5 reports the distribution of contribution ranks assigned across two scenarios: (i) three powerful agents backed by the Qwen-2.5-7B-Instruct model paired with one Qwen-2.5-1.5B-Instruct agent, and (ii) two agents of each type.

Table 3 summarizes AQUA-RAT performance under these settings. In case (i), where three strong and one weak agent are present, the single-agent performance is 71.65, while SELFORG raises it to 75.98, approaching the 76.77 level achieved when all four agents are strong. In case (ii), with two strong and two weak agents, SELFORG again yields large gains, improving accuracy from 66.54 in the single baseline to 74.80. These results demonstrate that SELFORG is able to reliably mitigate the drag introduced by weaker models, often recovering performance close to the all-strong setting.

Table 3: **Performance with weak agents in the pool (AQUA-RAT).** Comparison of SELFORG against single-agent baselines in the (3 strong vs 1 weak) and (2 strong vs 2 weak) settings.

| Method | $\mathcal{A}_1$ | $\mathcal{A}_2$ | $\mathcal{A}_3$ | $\mathcal{A}_4$ | AQUA-RAT | Note |
|---|---|---|---|---|---|---|
| Single | | | 1.5B | | 51.18 | Single agent; Qwen-1.5B backbone (single weak) |
| Single | | | 7B | | 76.77 | Single agent; Qwen-7B backbones (single strong) |
| Single | 7B | 7B | 7B | 1.5B | 71.65 | Backbone assignment is random per query (7B prob. 0.75) |
| SELFORG | 7B | 7B | 7B | 1.5B | 75.98 | Each agent uses its fixed backbone |
| Single | 7B | 7B | 1.5B | 1.5B | 66.54 | Backbone assignment is random per query (7B prob. 0.5) |
| SELFORG | 7B | 7B | 1.5B | 1.5B | 74.80 | Each agent uses its fixed backbone |

In setting (i), the weak agent is consistently identified as the least contributive, being placed in rank-4 in the majority of runs (68.1%). The stronger 7B models distribute across the higher ranks, demonstrating that the contribution estimation mechanism sharply separates weak from strong participants. The observation supports the theoretical guarantee in Section 2.6, namely that agreement

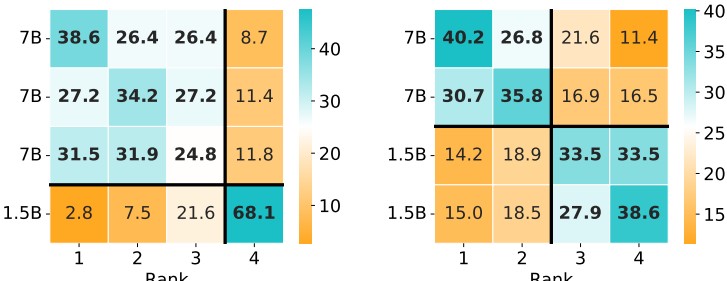

Figure 5: Heatmaps of ranking outcomes with a weak agent in the pool. Each heatmap depicts the percentage (%) of times agents were assigned to contribution ranks (rank 1 = highest contribution, rank 4 = weakest). The y-axis denotes the model type (Qwen-2.5-{7, 1.5}B-Instruct) assigned to each agent.

among correct agents amplifies their contribution scores, relegating weaker outliers downstream in the communication graph.

The case (ii) exhibits a more competitive dynamic. While the 1.5B agents remain overrepresented in the lower ranks, they also occasionally occupy intermediate positions (ranks 2 and 3), and the separation between strong and weak agents becomes less pronounced (due to the fact that the weak agents occasionally produce correct answers, thus leading to increased variability in contribution signals). Nevertheless, the stronger agents still dominate the top positions, ensuring that information flow in the communication graph is largely governed by higher-quality responses.

### D.2 TOKEN CONSUMPTION

We compare SELFORG to prior coordination frameworks with respect to both accuracy and token efficiency. Figures 6 and 7 visualize this trade-off, where bubble area corresponds to total token usage. For clarity, only DyLAN and MacNet are included among the baselines in the plots. Although AgentVerse and AutoGen achieve lower token usage than all other methods, their performance is substantially weaker (Table 1), with AutoGen in particular failing across nearly all benchmarks. Since our objective is to highlight the efficiency of coordination methods that remain competitive in accuracy, we restrict the visualization to DyLAN and MacNet.

By contrast, DyLAN and MacNet represent stronger baselines that consume a similar number of tokens as SELFORG. DyLAN exhibits relatively competitive performance on some reasoning tasks, but its overall average lags behind, especially on challenging datasets such as MMLU-Pro. MacNet shows modest efficiency advantages in prompt token usage but suffers from accuracy degradation across nearly all tasks. In both cases, SELFORG outperforms these baselines in accuracy while maintaining a comparable token budget, indicating a more favorable accuracy-efficiency trade-off.

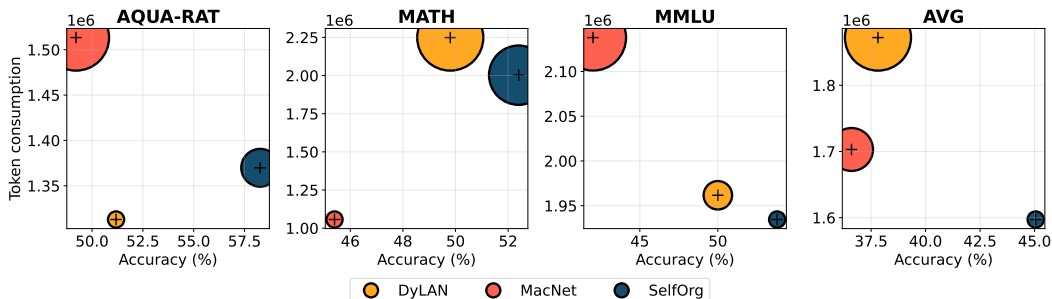

Figure 6: **Visualization of performance and completion token consumption.** Each bubble corresponds to a coordination method, with bubble area proportional to token consumption. Corresponding table: Table 1.

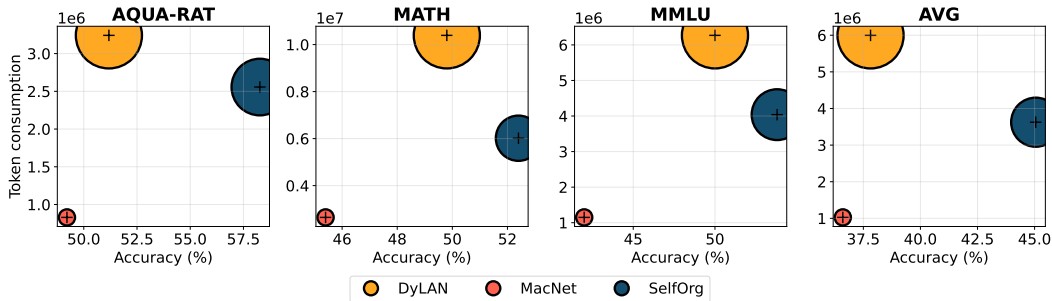

Figure 7: **Visualization of performance and prompt token consumption.** Each bubble corresponds to a coordination method, with bubble area proportional to token consumption. Corresponding table: Table 1.

Table 4: **Token consumption across coordination methods.** Completion tokens (top) and prompt tokens (bottom) consumed on each dataset on Qwen-2.5-1.5B-Instruct model. Corresponding table: Table 1.

| Method | MATH | GSM8K | AQUA-RAT | GSM-Hard | MMLU | MMLU-P | AIME |
|---|---|---|---|---|---|---|---|
| | | | completion tokens | | | | |
| DyLAN | 2249026 | 2086972 | 1312830 | 2468878 | 1961663 | 2786528 | 238078 |
| MacNet | 1056599 | 1806092 | 1513390 | 2238769 | 2137874 | 2925015 | 243205 |
| AgentVerse | 1077488 | 609241 | 530435 | 711561 | 338957 | 703302 | 74665 |
| AutoGen | 487744 | 282592 | 202713 | 371429 | 271488 | 390695 | 53990 |
| SELFORG | 2002530 | 1858577 | 1369879 | 2214019 | 1934568 | 1587246 | 213939 |

| Method | MATH | GSM8K | AQUA-RAT | GSM-Hard | MMLU | MMLU-P | AIME |
|---|---|---|---|---|---|---|---|
| | | | prompt tokens | | | | |
| DyLAN | 10391904 | 4706386 | 3241463 | 5719811 | 6267944 | 10847226 | 797505 |
| MacNet | 2647651 | 536500 | 829202 | 486320 | 1149266 | 1471736 | 61122 |
| AgentVerse | 3309868 | 2048995 | 1793383 | 2283561 | 1338962 | 2723973 | 240881 |
| AutoGen | 2026745 | 1292144 | 874703 | 1564267 | 1442236 | 2050001 | 176709 |
| SELFORG | 6016239 | 3836070 | 2556599 | 4351062 | 4038531 | 4251306 | 325588 |

### D.3 PROBABILISTIC MODELING. ADDITIONAL CORRELATION RESULTS

To further substantiate the relationship between embedding similarity and answer correctness, we extend the empirical analysis beyond the illustrative instance (Figure 2) and evaluate the phenomenon across three benchmarks: GSM-Hard, AQUA-RAT, and MMLU in Figure 8. Across

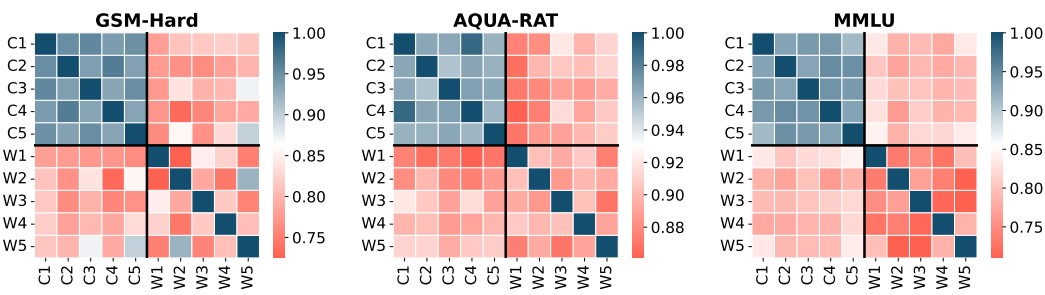

Figure 8: Cosine-similarity heatmaps of response embeddings (correct vs. incorrect responses) across three benchmarks.

Table 5: For randomly sampled queries, $p$ denotes the probability of generating the correct answer over 100 runs; $\max_k p_k$ is the largest probability of incorrect answers.

| | GSM-Hard | | AQUA-RAT | | MMLU | |
|---|---|---|---|---|---|---|
| | $p$ | $\max_k p_k$ | $p$ | $\max_k p_k$ | $p$ | $\max_k p_k$ |
| 1 | 0.98 | 0.01 | 0.60 | 0.11 | 0.61 | 0.14 |
| 2 | 0.35 | 0.05 | 0.93 | 0.02 | 0.57 | 0.16 |
| 3 | 0.74 | 0.02 | 0.90 | 0.04 | 0.42 | 0.20 |
| 4 | 0.71 | 0.03 | 0.84 | 0.06 | 0.75 | 0.10 |
| 5 | 0.39 | 0.09 | 0.23 | 0.21 | 0.42 | 0.21 |
| 6 | 0.31 | 0.07 | 0.40 | 0.16 | 0.80 | 0.07 |
| 7 | 0.70 | 0.04 | 0.70 | 0.08 | 0.84 | 0.07 |

benchmarks, we observe a consistent geometric signature in the response-embedding space: embeddings of correct answers form a tight cluster, whereas incorrect answers are comparatively dispersed, with low cross-similarity between the two groups. This structure persists broadly across both open-ended and multiple choice settings.

In all three cases, (i) correct responses exhibit high within-group similarity, (ii) incorrect responses are more weakly correlated with each other, and (iii) cross-block similarities remain low. This pattern is consistent with the modeling assumption that correct responses are semantically aligned, while incorrect responses scatter across error modes, implying that correct responders tend to align more with the population centroid than incorrect responders.

To complement the qualitative heatmap plots, we randomly sample multiple queries from each task and estimate (over repeated stochastic generations) the probability of producing the correct answer, denoted by $p$, and the maximum probability of the most frequently occurring incorrect response, denoted by $\max_k p_k$. Table 5 reports these values. Across GSM-Hard, AQUA-RAT, and MMLU, $p$ consistently exceeds $\max_k p_k$ by a substantial margin (satisfying the assumption in Lemma 2), indicating that correctness concentrates probability mass more strongly than any competing incorrect alternative.

Notably, the narrowest separation appears in the 5th AQUA-RAT example (highlighted in red), where $p \approx 0.23$ and $\max_k p_k \approx 0.21$. This case is consistent with near-random behavior: AQUA-RAT has five answer options, so uninformed guessing concentrates around $p \approx 0.2$, and no semantically coherent correct cluster is expected to dominate in such instances. Outside of these near-chance cases, the observed gaps support the mechanism underlying the contribution dominance.

## D.4  EMBEDDING MODEL

In our main experiments, we employ the `all-MiniLM-L6-v2` (Reimers & Gurevych, 2019) model, a lightweight embedding model with only 22.7M parameters, to estimate similarity between agent responses. This choice is intentional: we aim to keep the method efficient and avoid reliance on large embedding models, even if this introduces some additional noise into similarity estimates.

To validate this design choice, we conduct an ablation study in the *weak-agent-in-a-pool* scenario using different embedding models (see Figure 9). In addition to all-MiniLM, we evaluate all-MPNet-base-v2 (109M parameters) (Reimers & Gurevych, 2019) and Qwen3-0.6B-Embedding (600M parameters) (Zhang et al., 2025d). Across all cases, the embedding models are able to correctly identify the weakest agent: the weak participant is consistently ranked lowest in the majority of runs. Moreover, both MPNet and Qwen-0.6B provide sharper separation between strong and weak agents compared to MiniLM, reflecting their stronger representational capacity.

Nevertheless, our goal is to design a coordination mechanism that remains effective with lightweight embeddings. Despite the noisier similarity signals from all-MiniLM, SELFORG still succeeds in differentiating weak and strong contributors and delivers strong overall performance. This confirms that our approach does not require powerful encoders and can operate effectively under a minimal embedding budget, making it broadly applicable in resource-constrained settings.

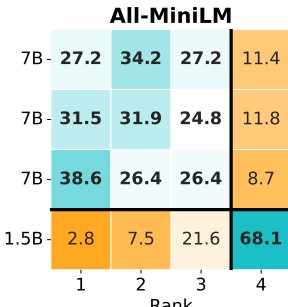 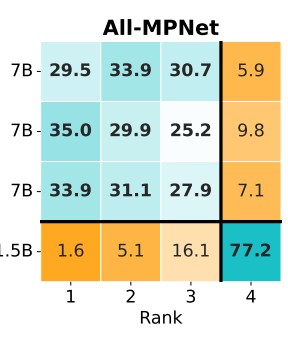 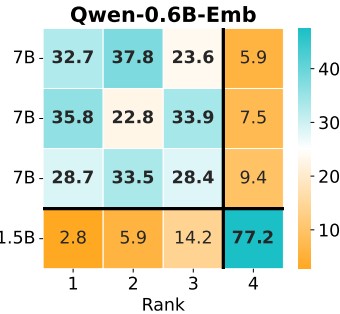

Figure 9: **Embedding model comparison in the weak-agent-in-a-pool scenario.** Heatmaps show the percentage of times of each agent (rows) being assigned to contribution ranks (columns) when using different embedding models for similarity estimation: `All-MiniLM` (22.7M parameters), `All-MPNet` (109M), and `Qwen-0.6B` (600M). All models are able to correctly identify the weakest agent ($\mathcal{A}_4$), with MPNet and Qwen-0.6B providing sharper separation between strong and weak agents.

## D.5 HETEROGENEOUS AGENTS WITH DYNAMIC ROLES

We additionally evaluate a dynamic role-switching variant of the heterogeneous-agent setting from Section 3.3 where agent roles are dynamic and not pre-fixed. Dynamic role settings yield a small improvement over fixed roles (accuracy: 66.93% vs. 66.14%, Figure 4), suggesting that contribution-guided routing remains effective without a static role prior and can achieve similar performance with permutation diversity.

We present Figure 10, which reports the percentage of times each agents attains contribution ranks ($1-4$). The distribution indicates a systematic separation between stronger and weaker agents: Qwen appears most frequently in earlier positions (rank 1), while Mistral is most frequently placed later (rank 4).

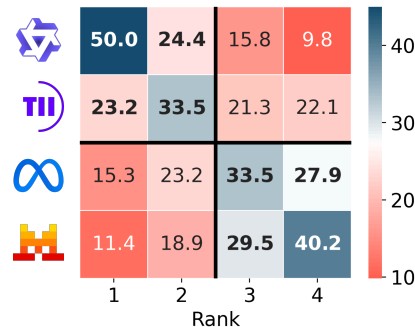

Figure 10: Dynamic role switching in SELFORG with heterogeneous agents.

## E EFFICIENT SELFORG

While the main pipeline of SELFORG proceeds through multiple rounds, not all rounds are equally necessary. In practice, if the agents already achieve strong agreement, further refinement may waste tokens without improving accuracy. To address this, we introduce an **early-stopping mechanism** based on natural consensus among peers.

**Consensus Criterion.** Let the similarity matrix $\mathbf{S} \in [-1, 1]$ be defined as in Section 2.4, where $\mathbf{S}_{n,m} = \cos(\mathbf{r}_n, \mathbf{r}_m)$ encodes the pairwise agreement between agents $n$ and $m$. We define the *minimum consensus* across all pairs as $\mathbf{S}_{\min} = \min_{n \neq m} \mathbf{S}_{n,m}$. Intuitively, $\mathbf{S}_{\min}$ captures the weakest agreement within the system. If this minimum exceeds a predefined threshold $\gamma \in [0, 1]$, then the agents are deemed to have reached sufficient consensus.

Formally, the system halts further rounds if $\mathbf{S}_{\min} \geq \gamma$, where $\gamma$ is the *consensus parameter* controlling strictness of agreement. For example, $\gamma = 0.9$ requires that all pairs of responses have at least 90% cosine similarity. When satisfied, the system outputs the centroid-based final response (Equation 6) without additional rounds.

This mechanism directly builds upon the communication graph formation step (Section 2.4). Since embeddings and similarities are already computed, evaluating $\mathbf{S}_{\min}$ incurs negligible overhead. By stopping once consensus is achieved, SELFORG avoids redundant propagation and aggregation, yielding substantial *token efficiency*. In scenarios where weak agents initially diverge, multiple

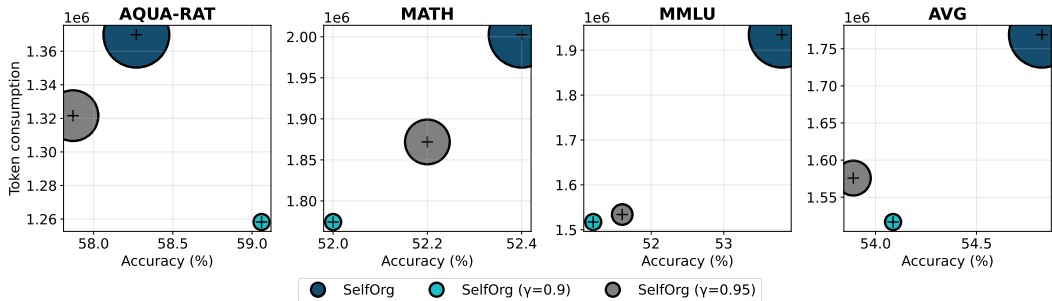

Figure 11: **Visualization of performance and completion token consumption** across benchmarks (AQUA-RAT, MATH, MMLU, and overall average). Each point corresponds to a method, with bubble size proportional to token usage. Methods include original SELFORG and efficient SELFORG with early stopping at $\gamma = \{0.9, 0.95\}$. Early stopping variants show improved efficiency (fewer tokens) while maintaining comparable accuracy.

rounds remain valuable; however, when natural agreement arises early, Efficient SELFORG prevents unnecessary computation.

**Experimental Results.** Figure 11 compares the baseline SELFORG with its early-stopping variants under consensus thresholds $\gamma \in \{0.9, 0.95\}$ on AQUA-RAT, MATH, and MMLU. All experiments were run with $N = 4$ agents, each selecting its top-2 neighbors, and 3 rounds. We report both accuracy and completion token consumption. Bubble sizes reflect token usage, with smaller bubbles denoting higher efficiency.

Baseline SELFORG achieves accuracies of $58.27\%$ (AQUA-RAT), $52.40\%$ (MATH), and $53.80\%$ (MMLU). Under $\gamma = 0.95$, accuracy slightly drops on AQUA-RAT ($57.87\%$), MMLU ($51.60\%$), and MATH ($52.2\%$). With a looser threshold $\gamma = 0.9$, performance closely matches or even exceeds the baseline on AQUA-RAT ($59.06\%$), while remaining comparable on MATH ($52.00\%$) and MMLU ($51.20\%$). This indicates that early stopping preserves task quality and, in some cases, improves it by preventing over-refinement.

The key advantage lies in efficiency. Both early-stopping settings consistently reduce token usage compared to the baseline. The stricter $\gamma = 0.95$ yields moderate savings, while the looser $\gamma = 0.9$ achieves the largest reductions. In relative terms, token usage decreases substantially while accuracy remains stable, with savings on the order of $10 - 15\%$ across benchmarks.

**Summary.** Efficient SELFORG demonstrates that natural peer consensus can serve as a reliable early-stopping signal. By halting once strong agreement is reached, the system avoids redundant message-passing rounds, improving token efficiency while preserving accuracy. Unlike prior MAS

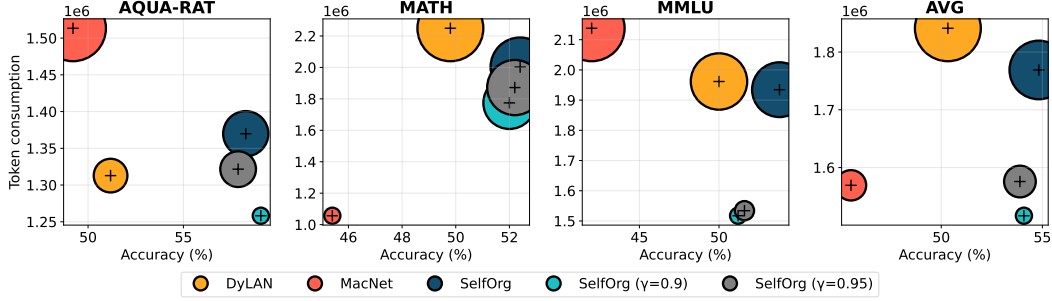

Figure 12: **Visualization of performance and completion token consumption** across benchmarks (AQUA-RAT, MATH, MMLU, and overall average). Each point corresponds to a method, with bubble size proportional to token usage. Methods include DyLAN, MacNet, SELFORG and efficient SELFORG with early stopping at $\gamma = \{0.9, 0.95\}$.

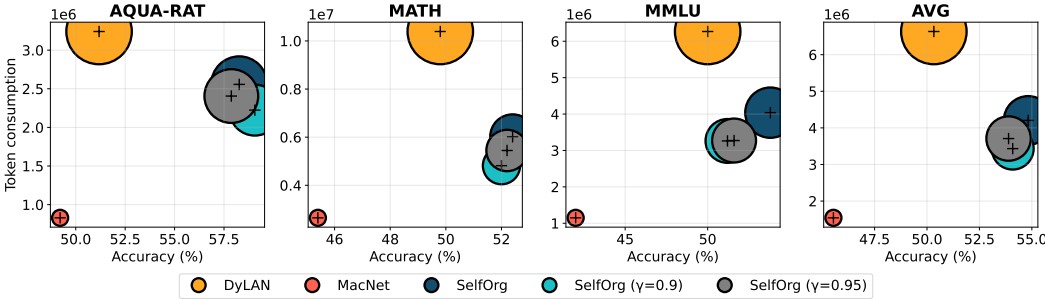

Figure 13: **Visualization of performance and prompt token consumption** across benchmarks (AQUA-RAT, MATH, MMLU, and overall average). Each point corresponds to a method, with bubble size proportional to token usage. Methods include DyLAN, MacNet, SELFORG and efficient SELFORG with early stopping at $\gamma = \{0.9, 0.95\}$.

approaches such as DyLAN, which require *explicit answer extraction* from responses to measure consensus (and may fail if the LLM deviates from formatting instructions), our method operates purely in the embedding space and thus avoids brittle dependencies on response parsing. Similarly, works that rely on external LLM judges to check consensus introduce additional computational and monetary overhead. In contrast, Efficient SELFORG is lightweight, model-agnostic, and robust: no answer extraction is needed, no external judge is invoked, and consensus is measured semantically rather than syntactically. This makes it especially suitable for scaling to large agent pools and diverse task domains.

For completeness, we provide Figures 12 and 13, which include efficient SELFORG along with the other baseline methods and depict the performance and completion/prompt token consumption.

### E.1 TERMINATION BREAKDOWN.

Efficient SELFORG halts in one of two ways: (i) it reaches the pre-specified maximum number of communication rounds, or (ii) it stops early once semantic consensus is achieved, i.e., when the minimum pairwise cosine similarity among agent responses satisfies $\mathbf{S}_{\min} \geq \gamma$ (consensus threshold; as described above) before exhausting the round budget.

Table 6 reports the breakdown across datasets under two thresholds. With the looser consensus requirement $\gamma = 0.9$, roughly $30-39\%$ of sessions terminate early; tightening the threshold to $\gamma = 0.95$ reduces early terminations to about $11-15\%$. This behavior is consistent with the role of $\gamma$ as an efficiency parameter: stricter value of $\gamma$ demands stronger agreement (fewer early stops), while looser values trigger early stopping more often, reducing token usage while keeping accuracy comparable in aggregate.

Table 6: Fraction of runs terminating by round limit ("All rounds") versus early stopping due to consensus ($\gamma$) under two thresholds.

| $\gamma$ | $\gamma = 0.9$ | | $\gamma = 0.95$ | |
|---|---|---|---|---|
| **Dataset** | **All rounds** | **Early stopped** | **All rounds** | **Early stopped** |
| AQUA-RAT | 0.669 | 0.331 | 0.846 | 0.154 |
| MATH | 0.612 | 0.388 | 0.852 | 0.148 |
| MMLU | 0.700 | 0.300 | 0.894 | 0.106 |

## F ABLATION STUDY

### F.1 NUMBER OF AGENTS

We conduct an ablation study to analyze the effect of the number of agents on both accuracy and efficiency. Figure 14 reports results for Qwen-2.5-1.5B-Instruct on the AQUA-RAT benchmark. The

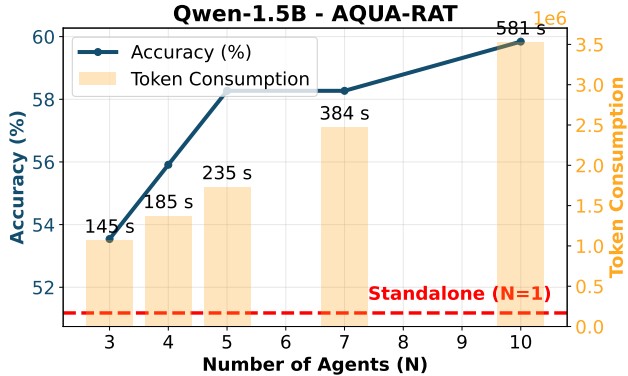

Figure 14: **Ablation on the number of agents.** Results for Qwen-2.5-1.5B-Instruct on AQUA-RAT. The blue line (left axis) shows accuracy as the number of agents $N$ increases, while orange bars (right axis) show token consumption. Latency (s) is annotated above each bar. Accuracy improves with more agents, but at the cost of higher latency and token usage, illustrating the trade-off between performance and efficiency in multi-agent coordination.

left $y$-axis shows accuracy, while the right $y$-axis shows token consumption; latency (in seconds) is annotated above each bar.

We observe that increasing the number of agents improves accuracy, from $53.54\%$ with $N = 3$ agents to $59.84\%$ with $N = 10$. However, this gain comes at the cost of both higher token usage (scaling from $1.07M$ to $3.53M$ tokens) and longer latency (from 145s to 581s). Interestingly, accuracy improvements are not strictly monotonic with $N$: performance plateaus at $58.27\%$ for $N = 5$ and $N = 7$, before rising again at $N = 10$. This suggests diminishing returns when adding additional weak agents, with benefits re-emerging only when coordination capacity (via $K$) increases sufficiently.

Overall, the ablation highlights the trade-off between accuracy and efficiency: more agents improve reliability but induce significant computational overhead, pointing to the importance of balancing scale against efficiency in multi-agent design.

## F.2    TO REFORM OR NOT TO REFORM

An important design choice in SELFORG is whether to *reform* the communication graph between rounds of interaction. Reforming allows agents to dynamically update their information flow structure based on the latest responses, while a static graph keeps the initial topology fixed throughout. We conduct an ablation study on two benchmarks, GSM8K and MMLU, using $N = 5$ agents and a neighbor budget of $K = 3$, to evaluate the impact of graph reform.

Table 7: Ablation on reforming the communication graph across rounds.

| Dataset | Reform | N | K | Accuracy |
|---------|--------|---|---|----------|
| GSM8K | True | 5 | 3 | 73.8 |
|         | False | 5 | 3 | 73.2 |
| MMLU | True | 5 | 3 | 52.8 |
|        | False | 5 | 3 | 51.4 |

As shown in Table 7, reforming the graph consistently improves performance, though the absolute gains are modest. This suggests that while the initial communication structure already captures useful alignment among agents, dynamically restructuring the graph allows the system to consolidate correct signals more effectively, especially on more challenging knowledge-intensive tasks. The relatively small gap also indicates that SELFORG is robust to whether reform is applied but benefits from it most in settings where agent responses are more diverse and noisy.

## F.3 AGGREGATION STRATEGIES

We evaluate the sensitivity of SELFORG to the final-answer aggregation rule by substituting several alternative methods while keeping the rest of the method unchanged. Table 8 reports the performance on MMLU over 3 runs. The contribution-weighted centroid used in SELFORG is competitive with selecting the highest contributor across all rounds, and both outperform the uniform centroid and voting-based baselines. Naive majority voting suffers a large drop and high variance due to option-letter parsing errors; using an LLM to adjudicate the majority recovers most of this gap but introduces additional variance.

Table 8: Ablation study on the aggregation methods on MMLU (mean $\pm$ std over 3 runs). *Naive majority voting over extracted option letters; parsing errors materially degrade performance and increase variance.

| Aggregation method | MMLU [mean $\pm$ std] |
|---|---|
| Contribution-weighted centroid (SELFORG , default) | **53.60 $\pm$ 1.27** |
| Uniform centroid | 52.70 $\pm$ 1.31 |
| Majority voting* | 46.17 $\pm$ 5.35 |
| Majority voting (with LLM deciding) | 51.90 $\pm$ 2.33 |
| Highest contributor across all rounds | 53.20 $\pm$ 1.19 |

## F.4 SELFORG WITH MEMORY AND TOOLS

All main-table results are reported in a tool-agnostic setting to isolate orchestration quality. Here we show that SELFORG's response-conditioned routing remains effective when agents are augmented with (i) per-agent memory (a scratchpad persisted across rounds) and (ii) a lightweight tool (sandboxed Python REPL for arithmetic checks; no external access). The orchestration mechanism is unchanged: contributions are still computed from the semantic content of agent responses and used to form the response-conditioned DAG.

Table 9: Effect of memory and tools on SELFORG with Qwen-1.5B backbone.

| Method (Qwen-1.5B) | GSM-Hard | MATH |
|---|---|---|
| Single | 36.2 | 49.2 |
| Self-Consistency | 37.2 | 50.2 |
| SELFORG (base; no memory/tools) | 38.0 | 52.4 |
| SELFORG + memory | 37.8 | 53.4 |
| SELFORG + memory + tools | **40.0** | **54.2** |

Table 9 compares single call, self-consistency (Wang et al., 2023), base SELFORG , and SELFORG with memory/tools under Qwen-1.5B. Base SELFORG improves upon both single and self-consistency methods. Adding memory yields comparable performance on GSM-Hard and a small gain on MATH, while adding both memory and tools provides the largest gains on both tasks.

## G  ADDITIONAL CODING EXPERIMENTS

To evaluate whether SELFORG extends beyond AQ/reasoning benchmarks, we add two coding evaluations using a weak backbone (Qwen-1.5B), where orchestration effects are expected to be most pronounced. We report HumanEval (Chen et al., 2021) with execution-based pass@1 and (ii) SRDD (Qian et al., 2024b) using the benchmark's composite score (completeness, executability, consistency) since SRDD has no single ground-truth solution.

Table 10: Collaborative coding results with Qwen-1.5B backbone model (mean $\pm$ std). HumanEval is pass@1; SRDD is the benchmark composite score.

| Method | HumanEval | SRDD |
|---:|:---:|:---:|
| Single | $47.31 \pm 0.91$ | $55.40 \pm 0.72$ |
| CoT | $46.34 \pm 0.75$ | $55.07 \pm 0.76$ |
| Self-Consistency | $45.63 \pm 1.41$ | $53.33 \pm 0.76$ |
| DyLAN | $42.37 \pm 3.47$ | $37.40 \pm 0.80$ |
| MacNet | $44.67 \pm 0.80$ | $56.00 \pm 0.40$ |
| SELFORG | $\mathbf{48.97 \pm 0.89}$ | $\mathbf{60.13 \pm 0.90}$ |

SELFORG is best on both HumanEval and SRDD, indicating that its contribution-guided communication improves not only reasoning but also collaborative coding.

