# OpenReview forum: "Stochastic Self-Organization in Multi-Agent Systems"
_ICLR.cc/2026/Conference — ICLR 2026 Poster_

### Official Review · Reviewer_gZmw · 2025-10-16

**Soundness:** 3
**Presentation:** 4
**Contribution:** 3
**Rating:** 6
**Confidence:** 3

**Summary:**

This paper proposes using response embeddings and similarity to the group opinion as a self-supervised signal to compute an approximate shapley value for attribution, and then constructing a graph heuristically based on node contributions to achieve better answer aggregation.

**Strengths:**

1. The paper’s motivation is well organized; a response-level, on-the-fly self-organizing topology for multi-agent systems is indeed compelling.
2. The overall methodological chain is clear and reasonable. Obtain signals in a self-supervised way, then use graph construction to optimize aggregation of those signals. There is theoretical support, and the approach aligns with intuition.

**Weaknesses:**

1. I think the premise “the correct answer appears repeatedly, while wrong answers scatter” only holds when the gap between model capability and task difficulty is not large. In that regime, SELFORG can be viewed as an improved self-consistency method.
2. When tasks are harder (e.g., on difficult mathematical reasoning where pass@1024 is low), correct responses are more likely to be a minority. In contrast, wrong responses may become the center of a conformity cluster. In that case, your self-supervised conformity signal may drown out the correct signal, and I don’t believe multiple rounds will reverse this failure mode.
3. There are many heuristic parameters; various thresholds and hyperparameters require manual tuning.
4. The graph construction is overly heuristic. Although the topology is not hard-coded, it is still generated by rules based on node contributions, which is insufficient to claim true emergence or self-organization.

**Questions:**

1. Have the authors tried other tasks? Currently, the datasets appear to be the common ones that have already been heavily overfit and are prone to data contamination.
2. Have you compared against baselines such as self-consistency? As it stands, this looks less like a multi-agent task setting and more like aggregating multiple responses. A single node lacks memory, tool use, and multi-turn interaction, and nodes may differ only by LLM or prompt.

---

> ### Author Response · Authors · 2025-11-21
> **Response to Reviewer gZmw [1/3]**
>
> We sincerely thank the reviewer for their positive evaluation, thoughtful assessment, and for highlighting key points that helped us clarify scope and robustness. We address each question below.
>
> > W1. I think the premise “the correct answer appears repeatedly, while wrong answers scatter” only holds when the gap between model capability and task difficulty is not large. In that regime, SELFORG can be viewed as an improved self-consistency method.
>
> The assumption that "correct answers appear repeatedly, while wrong answers scatter" is not unreasonable.  In fact,  a similar assumption forms the basis of any classifier ensemble. It is well-known that classifier ensembles work only when the predictions of individual classifiers are better than random (which leads to clustering of correct predictions) and the errors made by individual classifiers are somewhat uncorrelated (which leads to scattering of wrong predictions). A similar phenomenon is at play in multi-agent systems. Only when the gap between model capability and task difficulty is not large,  the responses of individual agents are better than random.  This is a core requirement for any collaboration to be meaningful.
>
> We agree that a non-degenerate competence level is necessary: our analysis assumes each agent answers correctly with probability $p > \text{random chance}$. If $p$ is a random chance (e.g., $\approx 0.25$ on a 4-way MCQ), no orchestration work can reliably recover the truth from a pool of answers. Our goal is coordination when the backbone model is capable of solving tasks with $p > \text{random chance} $.
>
> That said, SelfOrg is not simply self-consistency:
>
> - Self-consistency samples independent outputs from a single generator and picks a mode; agents do not revise their answers. SelfOrg performs round-level, response-conditioned message passing over a DAG that is rebuilt each round, letting agents update their responses using high-contribution peers. When the DAG degenerates to a single round with no edges, SelfOrg reduces to “vote-like” behavior; otherwise, it is strictly more expressive.
> - SelfOrg naturally supports heterogeneous agents, which decorrelates failure modes. Self-consistency samples from the same distribution (we now provide results for self-consistency as a baseline; please see Q2). Our contribution estimation, combined with DAG orientation, is designed to identify and propagate the correct cluster when $p$ adheres to the assumption.
> - So, they are different in (i) what gets aggregated, (ii) how information flows, and (iii) what capabilities the agents can attain (refer to Q2).
>
> > W2. When tasks are harder (e.g., on difficult mathematical reasoning where pass@1024 is low), correct responses are more likely to be a minority. In contrast, wrong responses may become the center of a conformity cluster. In that case, your self-supervised conformity signal may drown out the correct signal, and I don’t believe multiple rounds will reverse this failure mode.
>
> This highlights an important boundary condition that **all** orchestration methods face. Our assumption (stated in the paper) is that each agent answers correctly with probability $p$. For example, for a 4-way MCQ, if $p \approx 0.25$ (random guessing), no aggregation can reliably obtain the correct answer from a pool of samples. In such regimes, the issue is foundation model competence, not orchestration; a conformity cluster around a wrong answer indicates the base model is biased/misinformed. We do **not** claim SelfOrg corrects an **incompetent** backbone.
>
> In the intended regime (weak but capable backbones, i.e., $p > \text{random chance}$), our theoretical and empirical results apply: correct responses tend to cluster and receive higher contributions, so the DAG routes information from stronger to weaker agents. That is precisely where SelfOrg provides the largest gains: with Qwen-1.5B, we outperform other MAS baselines across QA, reasoning, and coding.

---

> ### Author Response · Authors · 2025-11-21
> **Response to Reviewer gZmw [2/3]**
>
> > W3. There are many heuristic parameters; various thresholds and hyperparameters require manual tuning.
>
> We respectfully disagree with the given statement. SelfOrg keeps parameters minimal and uses task-agnostic defaults:
>
> - Cosine similarity threshold $\tau$ (geometric alignment) [the only parameter SelfOrg uses]: fixed at $\tau=0.5$ for all 8 benchmarks (plus 2 new) and all backbones (10 different LLMs). **Ablation:** removing $\tau$ (keeping neighbors before the DAG enforcement step) yields similar accuracy with slightly higher prompt token consumption. So $\tau$ primarily improves efficiency, not correctness.
>
> - Rounds/Early Stopping: We use $T=3$ rounds by default. The optional consensus $\gamma \in \{0.9, 0.95\}$ (Efficient-SelfOrg only) saves token with negligible accuracy drop; e.g., early-stop fractions are $\approx 30-39$% ($\gamma = 0.9$) and $11-15$% ($\gamma=0.95$) on AQUA-RAT/MATH/MMLU.
>
> | method           | MATH   | GSM8K | AQUA  | GSM-H | MMLU  | MMLU-P | AIME  |
> |------------------|--------|-------|-------|-------|-------|--------|-------|
> | selforg          | 52.40  | 74.60 | 58.27 | 38.00 | 53.80 | 31.60  | 6.67  |
> | selforg (no tau) | 52.00  | 75.40 | 57.09 | 37.80 | 52.80 | 31.80  | 10.00 |
>
> Percentage increase in the (prompt token consumption) compared to the base SelfOrg
>
> | method           | MATH   | GSM8K | AQUA  | GSM-H | MMLU  | MMLU-P | AIME  |
> |------------------|--------|-------|-------|-------|-------|--------|-------|
> | selforg (no tau) | +3.56% | +5.25% | +4.94% | +4.45% | +5.99% | +0.99% | +4.00% |
>
> The usage of the completion token remains roughly the same; it does not increase.
>
> Percentage increase in the (total token consumption) compared to the base SelfOrg
>
> | method           | MATH   | GSM8K | AQUA  | GSM-H | MMLU  | MMLU-P | AIME  |
> |------------------|--------|-------|-------|-------|-------|--------|-------|
> | selforg (no tau) | +2.14% | +3.56% | +3.33% | +2.61% | +3.94% | +0.70% | +2.92% |
>
> In short, these parameters primarily manage budget/latency, rather than final accuracy.
>
> > W4. The graph construction is overly heuristic. Although the topology is not hard-coded, it is still generated by rules based on node contributions, which is insufficient to claim true emergence or self-organization.
>
> Our design is **principled** rather than ad hoc: edges and ranks derive from a Shapley-inspired contribution estimation with an $O(N)$ approximation, including bounds and ranking stability guarantees. The DAG is then formed by local, deterministic rules ($\tau$ similarity filtering $\to$ contribution ranking $\to$ cycle breaking to form the DAG structure).
>
> We use “self-organization” in the standard MAS sense: the topology emerges from local signals (the agents’ own responses), without pretraining a controller, without a judge LLM, and without RL over edges. Intuitively, placing high contributors upstream increases the probability that correct evidence propagates, precisely what we achieve under the stated assumption and observe empirically.
>
> > Q1. Have the authors tried other tasks? Currently, the datasets appear to be the common ones that have already been heavily overfit and are prone to data contamination.
>
> Yes. We added collaborative coding (SRDD) and HumanEval (execution-based pass@1) with Qwen-1.5B:
>
> | Method           | HumanEval    | SRDD          |
> |------------------|--------------|---------------|
> | Single           | 47.31 ± 0.91 | 55.40 ± 0.72  |
> | CoT              | 46.34 ± 0.75 | 55.07 ± 0.76  |
> | Self Consistency | 45.63 ± 1.41 | 53.33 ± 0.76  |
> | DyLAN            | 42.37 ± 3.47 | 37.40 ± 0.80  |
> | MacNet           | 44.67 ± 0.80 | 56.00 ± 0.40  |
> | SelfOrg          | **48.97 ± 0.89** | **60.13 ± 0.90**  |
>
> We also include AIME-2024 and GSM-Hard, which are widely used to reduce contamination risks. Results consistently favor SelfOrg in most of the regimes.

---

> ### Author Response · Authors · 2025-11-21
> **Response to Reviewer gZmw [3/3]**
>
> > Q2. Have you compared against baselines such as self-consistency? As it stands, this looks less like a multi-agent task setting and more like aggregating multiple responses. A single node lacks memory, tool use, and multi-turn interaction, and nodes may differ only by LLM or prompt.
>
> We thank the reviewer for this question.
>
> Self-consistency: Included above for HumanEval and SRDD. SelfOrg is superior to self-consistency with the same backbone. This supports our claim that response-conditioned orchestration adds value beyond sampling-and-vote strategies.
>
> Multi-turn: SelfOrg is explicitly multi-round (default $T=3$), and we provide an early-stopping variant.
>
> Memory/tools (orthogonal, but compatible): Our main results are tool-agnostic to keep comparisons clean and fair. Nevertheless, we conducted an ablation with Qwen-1.5B to address the reviewer’s question directly. We add the following capabilities to the agents: (i) memory (each agent carries a per-agent scratchpad across rounds) and (ii) a simple tool (Python REPL, with a properly sandboxed execution environment) for arithmetic checks (no external web).
> SelfOrg’s orchestration is unchanged: it still routes the information via the contribution-guided DAG using semantic embeddings of the agents’ responses, so it naturally extends to settings with memory/tool.
>
> | Method (Qwen-1.5B)               | GSM-Hard | MATH |
> |----------------------------------|----------|------|
> | Single                           | 36.2     | 49.2 |
> | Self-consistency                 | 37.2     | 50.2 |
> | SelfOrg (base) (no memory/tools) | 38       | 52.4 |
> | SelfOrg + memory                 | 37.8     | 53.4 |
> | SelfOrg + memory + tools         | 40       | 54.2 |
>
> On these tasks, we can see that base SelfOrg outperforms both single LLM call and self-consistency; adding memory helps gain a small performance on MATH (+1) and comparable performance on GSM-Hard (-0.2). Adding memory and tools gives the largest gains in both settings.
>
> We have conducted these new experiments, and SelfOrg continues to work in memory/tool settings because it only compares the semantic content of agent responses to compute contributions and routes information accordingly. For this paper, we keep the main tables tool-agnostic to ensure fairness and isolate orchestration quality. Moreover, exploring richer memory mechanisms and broader toolboxes is a promising direction, and we will pursue it in future work.

---

> > ### Comment · Reviewer_gZmw · 2025-11-26
> >
> > Thank you for the response, which resolves my concerns about the false premise and the hyperparameters. I will raise my Soundness score and will advocate for acceptance.

---

> > > ### Author Response · Authors · 2025-11-26
> > >
> > > We thank the reviewer for their acknowledgement of our rebuttal and for highlighting that we have addressed their concerns. We also thank the reviewer for increasing the soundness score from 3 (good) to 4 (excellent) and for their willingness to push for acceptance of our paper.
> > >
> > > Please let us know if any further clarification would be helpful.

---

### Official Review · Reviewer_SZuJ · 2025-10-31

**Soundness:** 3
**Presentation:** 4
**Contribution:** 3
**Rating:** 8
**Confidence:** 3

**Summary:**

The paper proposes a decentralized, response-conditioned multi-agent communication framework named SELFORG. Under the framework, agents first independently generate answers to the user query and use a Shapley-value–inspired approximation to assess peer contributions. A directed acyclic graph (DAG) is then constructed to govern information flow among agents, ensuring stable and efficient message transmission from high-contribution agents to others. Compared with prior approaches, SELFORG requires no pretrained topology generator, no edge-level reinforcement learning, and no external judge. The communication structure self-organizes round by round and is especially robust in the weak-backend regime. The paper further provides a theoretical analysis showing that increasing the number of agents raises the probability of obtaining the correct answer, and that correct responses naturally dominate the information flow, explaining the method’s effectiveness.

**Strengths:**

1. Unlike prior task-level and query-level communication frameworks, SELFORG performs round-level construction of the communication graph with low method complexity.
2. It uses a Shapley-inspired contribution estimate to build a DAG, placing high-contribution agents upstream to enable stable and efficient information propagation.
3. Both theory and experiments demonstrate the effectiveness of the proposed method.

**Weaknesses:**

1. The proposed method rests on the assumption that correct answers cluster more tightly in the embedding space while incorrect ones are more dispersed. The assumption may break down on tasks with multiple equivalent answer forms.

2. The construction of the communication graph relies on several heuristics—e.g., the similarity threshold $\tau$ and the number of neighbors $k$ per node—but the paper provides limited systematic robustness analysis of these hyperparameters.

3. The final output is selected as the response closest to the contribution-weighted centroid. This strategy may favor the dominant mode. When multiple valid solutions exist, the aggregation method may potentially suppresses minority—yet superior—reasoning paths. The paper does not provide comparisons with alternative aggregation methods like voting.

**Questions:**

See weaknesses.

**Details Of Ethics Concerns:**

The paper raises no ethical concerns.

---

> ### Author Response · Authors · 2025-11-21
> **Response to Reviewer SZuJ [1/2]**
>
> We sincerely thank the reviewer for the positive assessment and the constructive suggestions. We respond to the concerns below.
>
> > W1. The proposed method rests on the assumption that correct answers cluster more tightly in the embedding space while incorrect ones are more dispersed. The assumption may break down on tasks with multiple equivalent answer forms.
>
> We agree that tasks with many equivalent surface forms can challenge any approach that relies on representation geometry. Two clarifications:
>
> - Our contribution metric uses embedding similarity (cosine). This makes paraphrases and equivalent phrasings cluster naturally, a property we exploit when constructing the DAG.
> - We added the same correlation analysis across GSM-Hard, AQUA-RAT, and MMLU (refer to heatmaps: [https://ibb.co/HLjW0T9n](https://ibb.co/HLjW0T9n)). Quantitatively, for each dataset, we compare the probability that the response is correct ($p$) versus the best wrong cluster ($\max_i p_i$):
>
> | GSM-Hard | $p$    | $\max_i p_i$ |   | AQUA-RAT | $p$    | $\max_i p_i$ |   | MMLU | $p$    | $\max_i p_i$ |
> |----------|------|-----------|---|----------|------|-----------|---|------|------|-----------|
> | 1        | 0.98 | 0.01      |   | 1        | 0.6  | 0.11      |   | 1    | 0.61 | 0.14      |
> | 2        | 0.35 | 0.05      |   | 2        | 0.93 | 0.02      |   | 2    | 0.57 | 0.16      |
> | 3        | 0.74 | 0.02      |   | 3        | 0.9  | 0.04      |   | 3    | 0.42 | 0.2       |
> | 4        | 0.71 | 0.03      |   | 4        | 0.84 | 0.06      |   | 4    | 0.75 | 0.1       |
> | 5        | 0.39 | 0.09      |   | 5        | 0.23 | 0.21      |   | 5    | 0.42 | 0.21      |
> | 6        | 0.31 | 0.07      |   | 6        | 0.4  | 0.16      |   | 6    | 0.8  | 0.07      |
> | 7        | 0.7  | 0.04      |   | 7        | 0.7  | 0.08      |   | 7    | 0.84 | 0.07      |
>
> Across datasets, correct responses consistently dominate the wrong responses, supporting the modeling assumption that powers our contribution scores and DAG structure.
>
> **Edge cases.** For genuinely multi-modal correct outputs (e.g., several equally valid code designs) (expectedly, the semantics need to match, and SelfOrg is expected to work well as it operates in the embedding space), we note a simple safeguard that we can implement in the method: task-specific canonicalization (e.g., symbolic equivalence for algebra) prior to scoring.
>
>
> > W2. The construction of the communication graph relies on several heuristics—e.g., the similarity threshold $\tau$ and the number of neighbors $k$ per node—but the paper provides limited systematic robustness analysis of these hyperparameters.
>
> - $\tau$ (cosine-similarity threshold). We use a single geometric threshold $\tau$ that decides which peers an agent considers when forming edges. We fix $\tau=0.5 (60^{\circ})$ for all 8 benchmarks and all backbones. Importantly, $\tau$ is primarily an efficiency parameter: if we omit $\tau$ (i.e., consider all peers), accuracy is on par; thus, $\tau$ is not a fragile performance parameter.
> - $k$ (number of neighbors). In core SelfOrg, specifying $k$ is optional, as $k$ and $\tau$ play a similar role to serve efficiency, by filtering neighbors based on contributions (before pruning via cycle breaking and topological ordering). If we exclude $k$, accuracy is essentially unchanged (please see the experiment below where we do not use both $\tau$ and $k$). In other words, $k$ is purely for efficiency in prompt token consumption and is optional; removing it does not harm performance. We will make this explicit in the manuscript.
>
> | method           | MATH   | GSM8K | AQUA  | GSM-H | MMLU  | MMLU-P | AIME  |
> |------------------|--------|-------|-------|-------|-------|--------|-------|
> | selforg          | 52.40  | 74.60 | 58.27 | 38.00 | 53.80 | 31.60  | 6.67  |
> | selforg (no filter) | 52.00  | 75.40 | 57.09 | 37.80 | 52.80 | 31.80  | 10.00 |
>
> Percentage increase in the (prompt token consumption) compared to the base SelfOrg:
>
> | method           | MATH   | GSM8K | AQUA  | GSM-H | MMLU  | MMLU-P | AIME  |
> |------------------|--------|-------|-------|-------|-------|--------|-------|
> | selforg (no filter) | +3.56% | +5.25% | +4.94% | +4.45% | +5.99% | +0.99% | +4.00% |
>
> The usage of the completion token remains roughly the same; it does not increase.
>
> Percentage increase in the (total token consumption) compared to the base SelfOrg:
>
> | method           | MATH   | GSM8K | AQUA  | GSM-H | MMLU  | MMLU-P | AIME  |
> |------------------|--------|-------|-------|-------|-------|--------|-------|
> | selforg (no filter) | +2.14% | +3.56% | +3.33% | +2.61% | +3.94% | +0.70% | +2.92% |

---

> ### Author Response · Authors · 2025-11-21
> **Response to Reviewer SZuJ [2/2]**
>
> > W3. The final output is selected as the response closest to the contribution-weighted centroid. This strategy may favor the dominant mode. When multiple valid solutions exist, the aggregation method may potentially suppress minority—yet superior—reasoning paths. The paper does not provide comparisons with alternative aggregation methods, such as voting.
>
>
> We ran the reviewer-suggested alternatives and report mean ± std (3 runs) on MMLU:
>
> | Aggregation method                                                  | MMLU [mean ± std]  |
> |---------------------------------------------------------------------|--------------------|
> | Contribution-weighted centroid [ours]                               | 53.60 ± 1.27       |
> | Uniform centroid                                                    | 52.70 ± 1.31       |
> | Majority voting* (the reviewer suggested method)                    | 46.17 ± 5.35       |
> | Majority voting (with LLM deciding) (the reviewer suggested method) | 51.90 ± 2.33       |
> | Highest contributor across all rounds                               | 53.20 ± 1.19       |
>
> *Naive majority over-extracted letters suffers from parsing errors, which explains the larger drop.
>
>
> Because the response-conditioned DAG is robust, it amplifies high-contribution signals and suppresses noise, allowing agents to reach a stable consensus. So, results depend less on the precise aggregator. Our method and the reviewer's suggested method yield similar results [both outperform other methods]. Naive voting suffers from parsing errors; using an LLM to decide on the majority recovers most of this gap, but introduces a small additional variance, likely due to noise from this LLM.
>
> Regarding the reviewer’s concern about “minority but superior path”, we agree that this is important for tasks with multiple valid solutions. Since we operate in the embedding space, which works well with the semantic meaning of the responses, we expect that both correct results may lead to a similar embedding, thus reaching consensus. If this is not the case, a possible solution is to add a mode-aware fallback, where if the final responses exhibit high dispersion, we return the top-2 responses with the highest contribution scores (route both to the user). This could be explored in the future. Thanks for this question.

---

> > ### Author Response · Authors · 2025-11-27
> >
> > Thank you again for your thoughtful feedback. As the discussion period is ending soon, we would greatly appreciate hearing whether our rebuttal addressed your concerns, and we are happy to respond promptly to any additional comments.

---

> > > ### Comment · Reviewer_SZuJ · 2025-11-28
> > > **Thanks for the response**
> > >
> > > Thanks for providing additional experiments that resolves my concerns. I will keep my score and advocate for the acceptance.

---

### Official Review · Reviewer_F87g · 2025-11-01

**Soundness:** 3
**Presentation:** 3
**Contribution:** 2
**Rating:** 4
**Confidence:** 3

**Summary:**

This paper introduces SELFORG, a framework for orchestrating multi-agent systems (MAS) based on Large Language Models (LLMs) that dynamically constructs communication graphs without relying on external judges, pretrained topology generators, or reinforcement learning.

**Strengths:**

**Dynamic topology based on actual responses**: Unlike fixed graphs or query-specific topologies, SELFORG adapts to the stochastic agent outputs in real-time (Section 2.4), which is more realistic.

**Efficient Shapley value approximation**: Reduces complexity from O(2^N) to O(N) with theoretical bounds (Theorem 1) and ranking stability guarantees (Corollary 1).

**Strong results with weak models**: Achieves 45% accuracy with Qwen-1.5B vs 33-37% for other multi-agent methods (Table 1), proving the value of the approach when agents are unreliable.

**Weaknesses:**

**Limited to QA/reasoning tasks**: Only evaluates on MATH, GSM8K, MMLU etc. With strong models (Table 2), improvements are marginal (3-4%). No evaluation on genuine multi-agent tasks like MultiAgentBench or collaborative coding (SRDD).

**Missing key baseline**: MAS-GPT is mentioned (page 1) but never compared experimentally despite being the most similar approach.

**No statistical analysis**: Temperature=0.5 but no error bars or multiple runs reported. Only Figure 2 shows 100 runs for one problem.

**Strong assumptions**: Assumption 1 (correct answers cluster, wrong answers scatter) is only validated on one example (Figure 2b). All theory depends on this.

**Small scale only**: Tests use 4-5 agents. Scalability unclear.

**Questions:**

1. Why no comparison with "Who's the MVP? A Game-Theoretic Evaluation Benchmark for Modular Attribution in LLM Agents"? It also uses Shapley values for agent evaluation.

2. Algorithm 2 removes cycles arbitrarily - how sensitive is performance to different cycle removal orders?

3. How does this scale to 20-50 agents? The O(N²) similarity computation could be a bottleneck.

4. Section D.3: How is consensus threshold ξ chosen? Is it task-specific? What if wrong agents agree?

5. Why only test on tasks where single LLMs already work well? What about negotiation, collaborative design, or other inherently multi-agent problems? such as MultiAgentBench (Zhu et al. 2025)

6. The heterogeneous agent experiment (Section 3.3) uses fixed role assignment. What happens with dynamic role switching?

7. How many times was each experiment run? Standard deviations?

---

> ### Author Response · Authors · 2025-11-21
> **Response to Reviewer F87g [1/3]**
>
> We thank the reviewer for their time and constructive suggestions.
>
> > W1. Limited to QA/reasoning tasks: Only evaluates on MATH, GSM8K, MMLU etc. With strong models (Table 2), improvements are marginal (3-4%). No evaluation on genuine multi-agent tasks like MultiAgentBench or collaborative coding (SRDD).
>
> - On marginal gains (3-4%) with strong backbones. When the backbone model is already very strong, any orchestration layer has less headroom; single-agent accuracy is already high. In such regimes, even a 3-4% absolute marginal gain is meaningful. A 3-4% increase at ~65% accuracy corresponds to a ~11% error reduction, which is substantial. Moreover, our scaling-law summary shows that achieving even a 2% improvement via scaling alone typically requires doubling the model size (32B $\to$ 72B) (please see the result here, included for simplicity). In this context, the gain achieved by SelfOrg is not marginal.
> | AQUA-RAT (model size) | Accuracy (single call) |
> |-----------------------|------------------------|
> | 14B                   | 75.79                  |
> | 32B                   | 79.53                  |
> | 72B                   | 81.1                   |
>
> In contrast, SelfOrg’s gains are much larger where orchestration matters the most, with weak backbone models, because the collective can correct individual errors. (For example, with Qwen-1.5B, we are 8-12pp above other MAS baselines across benchmarks). This is precisely the gap SelfOrg is designed to close.
>
> - New evaluation on coding tasks. We now include SRDD collaborative coding and HumanEval results:
> | Method           | HumanEval    | SRDD          |
> |------------------|--------------|---------------|
> | Single           | 47.31 ± 0.91 | 55.40 ± 0.72  |
> | CoT              | 46.34 ± 0.75 | 55.07 ± 0.76  |
> | Self Consistency | 45.63 ± 1.41 | 53.33 ± 0.76  |
> | DyLAN            | 42.37 ± 3.47 | 37.40 ± 0.80  |
> | MacNet           | 44.67 ± 0.80 | 56.00 ± 0.40  |
> | SelfOrg          | **48.97 ± 0.89** | **60.13 ± 0.90**  |
>
> We use Qwen-1.5B as the backbone. On SRDD, we follow the benchmark’s composite score (completeness, executability, and consistency) since there is no ground truth; HumanEval is pass@1 with tests. SelfOrg is best on both, showing its benefits extend beyond reasoning to collaborative coding.
>
> > W2. Missing key baseline: MAS-GPT is mentioned (page 1) but never compared experimentally, despite being the most similar approach.
>
> We respectfully disagree with the point that MAS-GPT is the most similar approach to SelfOrg; in fact, MAS-GPT represents nearly the opposite design point. SelfOrg explicitly avoids the assumptions MAS-GPT relies on.
>
> - MAS-GPT trains a large LLM to orchestrate agents with task-specific instruction schemas (the dataset this model is trained on is basically the training set for these benchmarks); coupling that 32B controller with a 1.5B backbone is **infeasible** and also undermines the training-free goal.
> - Additionally, MAS-GPT adopts a different design point: the pretrained LLM (a 32B-sized model) selects an agent set and routes all candidate answers to a judge LLM, which then determines the final output; the agents can also invoke tools (e.g., code execution for mathematical operations). This pipeline is not comparable to our method, which is a training-free graph orchestration, and this work confounds results with (i) an extra trained model, (ii) a judge LLM, and (iii) tool availability. For that reason, we did not include MAS-GPT as a primary baseline in the main tables.
> - A closely related approach to MAS-GPT is G-Designer, which pretrains a query-conditioned graph generator. We did include G-Designer results in Table 1. It underperforms there, reinforcing our claim that pretrained generators do not help when underlying agents are weak, whereas SelfOrg, which adapts to actual responses at inference time, remains robust.
>
> For completeness, despite the methodological mismatch, we provide the extra experiments requested by the reviewer. Even with MAS-GPT’s pipeline, SelfOrg performs better in the weak-backend regime:
>
> | Method  | MATH | AQUA-RAT | MMLU | MMLU-Pro |
> |---------|------|----------|------|----------|
> | MAS-GPT | 49.6 | 53.15    | 47.8 | 25.2     |
> | SelfOrg | 52.4 | 58.27    | 53.8 | 31.6     |

---

> ### Author Response · Authors · 2025-11-21
> **Response to Reviewer F87g [2/3]**
>
> > W3. No statistical analysis: Temperature=0.5, but no error bars or multiple runs reported. Only Figure 2 shows 100 runs for one problem.
>
> Thank you for pointing this out. We have added multiple‑run results with standard deviations for the new coding tasks:
>
> | HumanEval        | Qwen-1.5B    | Qwen-14B     |
> |------------------|--------------|--------------|
> | Single           | 47.31 ± 0.91 | 80.49 ± 0.61 |
> | CoT              | 46.34 ± 0.75 | 81.51 ± 0.93 |
> | Self Consistency | 45.63 ± 1.41 | 77.91 ± 1.23 |
> | DyLAN            | 42.37 ± 3.47 | 74.23 ± 0.83 |
> | MacNet           | 44.67 ± 0.80 | 77.78 ± 0.67 |
> | SelfOrg          | 48.97 ± 0.89 | 81.40 ± 1.20 |
>
> These new multi-run coding results demonstrate statistical reliability. We commit to adding standard deviations in tables where the compute allows (e.g., with smaller-scale models) in the camera-ready version; but as can be seen from the standard deviations we include here, they are in the range with the baselines, and the average accuracy we achieve is much larger than the baselines.
>
> > W4. Strong assumptions: Assumption 1 (correct answers cluster, wrong answers scatter) is only validated on one example (Figure 2b). All theory depends on this.
>
> We extended the validation to both open-ended and multiple-choice tasks (GSM-Hard, AQUA-RAT, MMLU). The new heatmaps (attached) [https://ibb.co/HLjW0T9n](https://ibb.co/HLjW0T9n) show tight clustering among correct answers and dispersion among wrong ones. Quantitatively, for each dataset, we report the probability of getting the question correct ($p$) versus the probability of the best competing wrong answer ($\max_i p_i$). Correct clusters dominate in all three datasets, supporting the assumption that drives our contribution scores and the DAG’s behavior. This aligns with our theoretical results (Theorem 1, Corollary 1/2).
>
> | GSM-Hard | $p$    | $\max_i p_i$ |   | AQUA-RAT | $p$    | $\max_i p_i$ |   | MMLU | $p$    | $\max_i p_i$ |
> |----------|------|-----------|---|----------|------|-----------|---|------|------|-----------|
> | 1        | 0.98 | 0.01      |   | 1        | 0.6  | 0.11      |   | 1    | 0.61 | 0.14      |
> | 2        | 0.35 | 0.05      |   | 2        | 0.93 | 0.02      |   | 2    | 0.57 | 0.16      |
> | 3        | 0.74 | 0.02      |   | 3        | 0.9  | 0.04      |   | 3    | 0.42 | 0.2       |
> | 4        | 0.71 | 0.03      |   | 4        | 0.84 | 0.06      |   | 4    | 0.75 | 0.1       |
> | 5        | 0.39 | 0.09      |   | 5        | 0.23 | 0.21      |   | 5    | 0.42 | 0.21      |
> | 6        | 0.31 | 0.07      |   | 6        | 0.4  | 0.16      |   | 6    | 0.8  | 0.07      |
> | 7        | 0.7  | 0.04      |   | 7        | 0.7  | 0.08      |   | 7    | 0.84 | 0.07      |
>
> The assumption that "correct answers cluster and wrong answers scatter" is not unreasonable.  In fact, a similar assumption forms the basis of any classifier ensemble. It is well-known that classifier ensembles work only when the predictions of individual classifiers are better than random (which leads to clustering of correct predictions) and the errors made by individual classifiers are somewhat uncorrelated (which leads to scattering of wrong predictions). A similar phenomenon is at play in multi-agent systems.
>
> In the case of AQUA-RAT, 5-th result is simply a random guess, not a task that the agent can actually solve (since AQUA-RAT is a multiple-choice task and comprises 5 options, the random guess would concentrate around $p=0.2$).
>
> > W5. Small scale only: Tests use 4-5 agents. Scalability unclear.
>
> > Q3. Scaling to 20–50 agents; The O(N²) similarity computation could be a bottleneck.
>
> We would like to note that it is a norm for multi-agent collaboration works to assume this range of agents (4-5) [DyLAN, AgentVerse]; MacNet is a paper that studies scaling. We already have experiments in the paper where we scale $N$ up to 10 agents (please see Appendix E.1). Nevertheless, we ran the requested larger-$N$ study:
>
> | N  | acc   | latency (s) | token consumption (M) |
> |----|-------|-------------|-----------------------|
> | 20 | 59.06 | 892         | 4.8                   |
> | 50 | 59.45 | 1525        | 12.5                  |
>
> Accuracy **saturates** (similar to 10 agents) while latency/tokens grow roughly linearly with $N$. The pairwise similarity computation step is $O(N^2)$ but fast in practice (All-MiniLM embeddings (384-D); $N=50$ implies 1225 pairs, which is negligible compared to LLM calls with attention that scales quadratically with sequence length (1000+).

---

> ### Author Response · Authors · 2025-11-21
> **Response to Reviewer F87g [3/3]**
>
> > Q1. Why no comparison with “Who’s the MVP”?
>
> This work examines modular attribution using Shapley values to assess modules within an agent pipeline. It serves as an evaluation/attribution benchmark, rather than an orchestration method for multi-agent communication. In contrast, SelfOrg uses a Shapley value, $O(N)$ approximation online, to construct a communication DAG and weight edges among agents. We will cite this work as related attribution work, but it is not a baseline for comparison.
>
> > Q2. Algorithm 2 removes cycles arbitrarily - how sensitive is performance to different cycle removal orders?
>
> SelfOrg does not remove the cycles arbitrarily, but in the principal way. When a directed cycle is detected, we delete the edge from the lower-contribution agent to the higher-contribution agent. This rule is deterministic given the contribution ordering with a fixed tie-breaker and yields the same DAG for a given set of responses. Intuitively, information should not flow from weaker to stronger agents when resolving cycles; this improves robustness and prevents error amplification.
>
> > Q4. Section D.3: How is consensus threshold $\gamma$ chosen? Is it task-specific? What if wrong agents agree?
>
> Consensus threshold $\gamma$ is used only in Efficient-SelfOrg for early stopping. We study $\gamma \in \{0.9, 0.95\}$, which trades off compute for latency with minimal impact on accuracy (please see Appendix D.3). Breakdown:
>
> | gamma    | gamma = 0.9 |               | gamma = 0.95 |               |
> |----------|-------------|---------------|--------------|---------------|
> | dataset  | all rounds  | early stopped | all rounds   | early stopped |
> | AQUA-RAT | 0.669       | 0.331         | 0.846        | 0.154         |
> | MATH     | 0.612       | 0.388         | 0.852        | 0.148         |
> | MMLU     | 0.7         | 0.3           | 0.894        | 0.106         |
>
> Regarding consensus on wrong answers, it is indeed possible in principle, but is rare due to the dispersion of wrong answers (validated in new heatmaps); the DAG and contribution weights are designed to down-weight such cases.
>
> > Q5. Why only test on tasks where single LLMs already work well? What about negotiation, collaborative design, or other inherently multi-agent problems? Such as MultiAgentBench (Zhu et al. 2025)
>
> The QA suite we use is not straightforward; in fact, many MAS baselines underperform single-agent setting, especially with weak backbones (see Table 1 to observe how other methods collapse). That said, we added SRDD (collaborative coding) and HumanEval, and SelfOrg performs best in both. MultiAgentBench combines heavy tool use and environment control, which introduces confounds (such as tool availability and deterministic execution). SelfOrg is orthogonal to these and can be plugged into such settings; however, we focused on tool-free reasoning and coding for clear comparisons. We will cite MultiAgentBench in the paper.
>
> > Q6. The heterogeneous agent experiment (Section 3.3) uses fixed role assignment. What happens with dynamic role switching?
>
> We evaluated dynamic role switching and observed a slightly higher accuracy compared to fixed roles: 66.93 vs. 66.14. Contribution estimation continues to separate strong agents from weak ones. The order/role distribution is:
>
> | Order -> | 1      | 2      | 3      | 4      |
> |----------|--------|--------|--------|--------|
> | Qwen     | 0.5    | 0.2441 | 0.1575 | 0.0984 |
> | Falcon   | 0.2323 | 0.3346 | 0.2126 | 0.2205 |
> | LLaMA    | 0.1535 | 0.2323 | 0.3346 | 0.2795 |
> | Mistral  | 0.1142 | 0.189  | 0.2953 | 0.4016 |
>
> (We will include this as a heatmap in the revision.)
>
> > Q7. Standard deviations.
>
> Please refer to W3, where we report the results with three runs and standard deviation.

---

> > ### Author Response · Authors · 2025-11-27
> >
> > Thank you again for your thoughtful feedback. As the discussion period is ending soon, we would greatly appreciate hearing whether our rebuttal addressed your concerns, and we are happy to respond promptly to any additional comments.

---

### Official Review · Reviewer_TXyD · 2025-11-01

**Soundness:** 2
**Presentation:** 2
**Contribution:** 2
**Rating:** 4
**Confidence:** 4

**Summary:**

This paper proposes SELFORG, a training-free framework for stochastic self-organization in multi-agent systems that dynamically constructs communication graphs using an embedding-based contribution metric approximating Shapley values. By structuring information flow as a Directed Acyclic Graph to enhance valuable signals and suppress noise, SELFORG achieves consistent improvements on eight reasoning and QA benchmarks, especially when applied to weaker language models.

**Strengths:**

1. Simple, training-free mechanism that works across LLM sizes.
2. Novel approach to constructing the communication DAG by leveraging cosine similarity for its decision logic.
3. Extensive experiments across a wide array of reasoning benchmarks demonstrate consistent and significant improvements over strong baselines, especially when employing smaller-scale language models.

**Weaknesses:**

1. The framework's performance appears sensitive to key hyperparameters, such as the similarity threshold \tau and the termination variance. These variables may require tuning per dataset or model, which the paper does not address, leaving the impression that the chosen values are empirically derived heuristics rather than principled parameters.
2. The paper highlights that SELFORG is a lightweight framework requiring no pre-training or reinforcement learning, contrasting it with complex task-specific topologies. However, the token consumption table in the appendix shows that the method does not possess a significant advantage in resource usage over other baselines, particularly AutoGen.
3. The paper's claim of a positive correlation between the cosine similarity of LLM agent embeddings and answer correctness is supported by only a single heatmap from one case, which is not sufficiently convincing.

**Questions:**

1. Could you comment on the robustness of the chosen hyperparameters? Specifically, how sensitive is the model's performance to adjustments in the cosine similarity threshold and the maximum number of rounds?
2. Can you provide a breakdown of the termination triggers? Specifically, how frequently did the dialogue stop due to reaching the round limit versus achieving consensus based on the variance threshold?
3. The final answer is selected as the response with the highest contribution score in the very last round. Have you considered alternative aggregation methods, such as selecting the response with the highest contribution score across all rounds? Is there a risk that the collective converges on a high-agreement answer in the final round, while a more brilliant but initially less understood answer from an earlier round gets discarded?

---

> ### Author Response · Authors · 2025-11-21
> **Response to Reviewer TXyD [1/3]**
>
> We thank the reviewer for the careful reading and constructive suggestions. SelfOrg is a training-free, response-conditioned orchestration framework that forms a communication DAG using a Shapley value-based contribution estimation. Our implementation uses a fixed cosine threshold $\tau=0.5$ and a maximum of 3 rounds by default; an early-stopping consensus threshold $\gamma$ is used only in the efficiency variant (Efficient-SelfOrg). These settings and the aggregation rule are described in the paper.
>
> > W1/Q1. The framework's performance appears sensitive to key hyperparameters, such as the similarity threshold \tau and the termination variance. These variables may require tuning per dataset or model, which the paper does not address, leaving the impression that the chosen values are empirically derived heuristics rather than principled parameters.
>
> We respectfully disagree. Across all reported settings (8 benchmarks, 10 LLM backbone models, and multiple task types), a fixed $\tau=0.5$ suffices; while $\gamma$ is only for the efficient variant (i.e., not including $\gamma$ does not degrade performance) and, per Appendix D.3, does not make SelfOrg fragile.
>
> - Similarity threshold $\tau$. We only use one parameter for peer selection: cosine similarity threshold $\tau$, which has a geometric interpretation and not just any value, that decides which peers an agent communicates to when building the per-round DAG (Algorithm 2, lines 6-12 and lines 192-197). Intuitively, $\tau$ specifies the minimal directional alignment required for communication. We set $\tau=0.5 (=60^{\circ})$ once and use it unchanged for all 8 benchmarks [MATH, GSM-8K, AQUA-RAT, GSM-Hard, MMLU, MMLU-Pro, GPQA, AIME-2024] and across all 10 backbone models in the paper [Qwen2.5-{1.5B, 3B, 7B, 14B, 32B, 72B}, LLaMA3-{8B, 70B}, Falcon-7B, Mistral-7B]. Importantly, $\tau$ is not required for correctness: if $\tau$ is removed (no neighbor filtering before DAG formation), we observe on-par accuracy with higher token consumption; $\tau$ mainly serves efficiency purposes, similarly to $\gamma$ discussed next, by avoiding low-value edges.
>
> | method           | MATH   | GSM8K | AQUA  | GSM-H | MMLU  | MMLU-P | AIME  |
> |------------------|--------|-------|-------|-------|-------|--------|-------|
> | selforg          | 52.40  | 74.60 | 58.27 | 38.00 | 53.80 | 31.60  | 6.67  |
> | selforg (no tau) | 52.00  | 75.40 | 57.09 | 37.80 | 52.80 | 31.80  | 10.00 |
>
> Percentage increase in the (prompt token consumption) compared to the base SelfOrg:
>
> | method           | MATH   | GSM8K | AQUA  | GSM-H | MMLU  | MMLU-P | AIME  |
> |------------------|--------|-------|-------|-------|-------|--------|-------|
> | selforg (no tau) | +3.56% | +5.25% | +4.94% | +4.45% | +5.99% | +0.99% | +4.00% |
>
> The usage of the completion token remains roughly the same; it does not increase.
>
> Percentage increase in the (total token consumption) compared to the base SelfOrg:
>
> | method           | MATH   | GSM8K | AQUA  | GSM-H | MMLU  | MMLU-P | AIME  |
> |------------------|--------|-------|-------|-------|-------|--------|-------|
> | selforg (no tau) | +2.14% | +3.56% | +3.33% | +2.61% | +3.94% | +0.70% | +2.92% |
>
> - Early-stopping consensus threshold $\gamma$ (Efficient-SelfOrg only). The base SelfOrg does not utilize any termination variance; the efficient variant introduces $\gamma$ as an optional semantic-agreement criterion to end the dialogue early once agents have already agreed. Appendix D.3 shows $\gamma \in {0.9, 0.95}$ preserves accuracy while reducing tokens (details and exact numbers are in the paper). Thus $\gamma$ trades off compute for latency/efficiency and is orthogonal to correctness.
>
> - Number of rounds (T). We cap at $T=3$ rounds (including the decentralized initialization round). This choice aligns with our compute/latency budget (and is widely used in most baseline works, e.g., DyLAN, MacNet), and it exhibits diminishing returns after 2-3 refinements; this approach is also consistent with our findings throughout the paper.

---

> ### Author Response · Authors · 2025-11-21
> **Response to Reviewer TXyD [2/3]**
>
> > W2. The paper highlights that SELFORG is a lightweight framework requiring no pre-training or reinforcement learning, contrasting it with complex task-specific topologies. However, the token consumption table in the appendix shows that the method does not possess a significant advantage in resource usage over other baselines, particularly AutoGen.
>
> By **lightweight**, we refer to the orchestration cost (extra overhead in training + inference), not “fewest tokens”. SelfOrg requires no pre-training, no external judge LLM, and no RL over edges. The approaches that train graph generators, rely on judge LLMs, or optimize edges with RL, incur additional training/inference overhead that is orthogonal to tokens and absent in SelfOrg.
>
> For the competitive baselines, SelfOrg uses approximately the same or fewer inference tokens across benchmarks while achieving the best accuracy (see Appendix D.2). Autogen may use fewer tokens, but it is unusable because it is collapsing in the weak-backbone regime that our work targets, yielding the worst overall average (18.93) and ranking in Table 1.
>
> For efficient token usage, we provide Efficient SelfOrg ($\gamma \in \{0.9, 0.95\}$), which reduces tokens by $\approx 10-15$% while keeping accuracy on par, clarifying the efficiency story without changing the training-free nature of the framework.
>
> We apologize for the confusion caused, and we will clarify that the word lightweight means “training-free” and is not related to “lowest tokens”.
>
> > W3. The paper's claim of a positive correlation between the cosine similarity of LLM agent embeddings and answer correctness is supported by only a single heatmap from one case, which is not sufficiently convincing.
>
> We have expanded this analysis beyond a single case. The attached heatmaps (GSM-Hard, AQUA-RAT, MMLU) [https://ibb.co/HLjW0T9n](https://ibb.co/HLjW0T9n) display the same structure: correct answer embeddings cluster strongly with each other, wrong answers are relatively scattered, and cross-block similarities are low. This holds for both open-ended (GSM-Hard) and multiple-choice (AQUA-RAT, MMLU) settings.
>
> To provide further quantification, we randomly sample several queries from each dataset:
>
> | GSM-Hard | $p$    | $\max_i p_i$ |   | AQUA-RAT | $p$    | $\max_i p_i$ |   | MMLU | $p$    | $\max_i p_i$ |
> |----------|------|-----------|---|----------|------|-----------|---|------|------|-----------|
> | 1        | 0.98 | 0.01      |   | 1        | 0.6  | 0.11      |   | 1    | 0.61 | 0.14      |
> | 2        | 0.35 | 0.05      |   | 2        | 0.93 | 0.02      |   | 2    | 0.57 | 0.16      |
> | 3        | 0.74 | 0.02      |   | 3        | 0.9  | 0.04      |   | 3    | 0.42 | 0.2       |
> | 4        | 0.71 | 0.03      |   | 4        | 0.84 | 0.06      |   | 4    | 0.75 | 0.1       |
> | 5        | 0.39 | 0.09      |   | 5        | 0.23 | 0.21      |   | 5    | 0.42 | 0.21      |
> | 6        | 0.31 | 0.07      |   | 6        | 0.4  | 0.16      |   | 6    | 0.8  | 0.07      |
> | 7        | 0.7  | 0.04      |   | 7        | 0.7  | 0.08      |   | 7    | 0.84 | 0.07      |
>
> Across all three datasets, the probability of getting the question correct ($p$) dominates the best competing wrong answer ($\max_i p_i$); the gaps are large and consistent, with only narrow gaps in the AQUA-RAT 5-th example, where it is simply a random guess, not a task that the agent can actually solve (since AQUA-RAT is a multiple-choice task and comprises of 5 options, the random guess would concentrate around $p=0.2$). This directly supports the paper’s modeling, where correct answers form a tighter semantic cluster than wrong ones and therefore receive higher contribution scores.
>
> > Q2. Can you provide a breakdown of the termination triggers? Specifically, how frequently did the dialogue stop due to reaching the round limit versus achieving consensus based on the variance threshold?
>
> The requested breakdown is presented here:
>
> | gamma    | gamma = 0.9 |               | gamma = 0.95 |               |
> |----------|-------------|---------------|--------------|---------------|
> | dataset  | all rounds  | early stopped | all rounds   | early stopped |
> | AQUA-RAT | 0.669       | 0.331         | 0.846        | 0.154         |
> | MATH     | 0.612       | 0.388         | 0.852        | 0.148         |
> | MMLU     | 0.7         | 0.3           | 0.894        | 0.106         |
>
> So, $\gamma=0.9$ ends $\approx 30-39$% of sessions early; the stricter $\gamma=0.95$ ends $\approx 11-15$% early. This aligns with Appendix D.3: $\gamma$ modulates efficiency, keeping accuracy comparable while reducing token consumption.

---

> ### Author Response · Authors · 2025-11-21
> **Response to Reviewer TXyD [3/3]**
>
> > Q3. The final answer is selected as the response with the highest contribution score in the very last round. Have you considered alternative aggregation methods, such as selecting the response with the highest contribution score across all rounds? Is there a risk that the collective converges on a high-agreement answer in the final round, while a more brilliant but initially less understood answer from an earlier round gets discarded?
>
> Clarification. We do **not** pick “the response with the highest contribution score in the last round”. SelfOrg aggregates by a contribution-weighted centroid of the last-round embeddings and selects the observed response closest to that centroid (Eqs. 6-7).
>
> We implemented the suggested proposal by the reviewer and compared several other aggregation methods on MMLU:
>
> | Aggregation method                                                  | MMLU [mean ± std]  |
> |---------------------------------------------------------------------|--------------------|
> | Contribution-weighted centroid [ours]                               | 53.60 ± 1.27       |
> | Uniform centroid                                                    | 52.70 ± 1.31       |
> | Majority voting*                                                    | 46.17 ± 5.35       |
> | Majority voting (with LLM deciding)                                 | 51.90 ± 2.33       |
> | Highest contributor across all rounds (reviewer’s suggested method) | 53.20 ± 1.19       |
>
> *With simple majority voting over extracted letters (parsing errors hurt performance).
>
> We thank the reviewer for mentioning the importance of the aggregation rule. Because the communication formation is robust, it amplifies high-value signals and suppresses noisy edges, allowing agents to reach consensus; thus, performance depends much less on the specific aggregation rule. Empirically, our contribution-weighted centroid and the reviewer’s suggested method are essentially the same. Uniform centroid and majority voting are marginally below. Overall, the graph-based consensus is the main driver of accuracy.
>
> **Groupthink risk.** Yes, any collaborative method can converge on a high-agreement wrong answer. Our analysis in Section 2.6 explains why this is rare in practice under the observed dispersion of wrong answers: if wrong answers are scattered while correct ones cluster, then (i) agreement on the correct answer is more likely than agreement on the same wrong answer (Lemma 1), and (ii) correct responders receive strictly higher contribution scores (Lemma 2, Corollary 2). The new heatmaps and probability tables above empirically support these assumptions on multiple task types.
>
>
> We appreciate the reviewer’s suggestions, which helped us refine the presentation and incorporate new analyses.

---

> > ### Author Response · Authors · 2025-11-27
> >
> > Thank you again for your thoughtful feedback. As the discussion period is ending soon, we would greatly appreciate hearing whether our rebuttal addressed your concerns, and we are happy to respond promptly to any additional comments.

---

### Author Response · Authors · 2025-11-21
**General Response**

We sincerely thank the reviewers for their careful readings and for highlighting the core strengths of SelfOrg. Across reviews, there is a clear consensus on
1. our training-free, judge-free orchestration,
1. the response-conditioned construction of a DAG using Shapley-value-based contribution estimation,
1. the theoretical support and methodological clarity,
1. consistent gains, especially in the weak-backbone regime.

We also appreciate the concrete questions raised across reviews (robustness, aggregation, assumption validation, and additional tasks). Our rebuttal includes multi-dataset correlation analyses, aggregation ablations, SRDD/HumanEval coding experiments, and responses to reviewer-specific questions, covering all points raised.

---

### Author Response · Authors · 2025-12-03
**Summary of Reviewer Feedback and Rebuttal Clarifications**

We thank the Area Chair for stepping in during this unusual situation. Below, we summarize the main strengths as highlighted by reviewers and our rebuttal to their raised concerns.

---

Across the board, reviewers consistently highlighted core strengths of SelfOrg:

- First, all reviewers (**TXyD, F87g, SZuJ, gZmw**) emphasized the appeal of a **training-free, judge-free orchestration framework**. This was repeatedly praised as a major practical advantage.

- Second, all reviewers (**TXyD, F87g, SZuJ, gZmw**) pointed out that the response-conditioned DAG based on a Shapley-inspired contribution estimator is novel, principled, and effective.

- Third, reviewers (**F87g, SZuJ, gZmw**) explicitly commended the clarity and theoretical grounding of the method, noting that the results (e.g., stability guarantees, theorems) align well with intuition.

- Finally, all reviewers (**TXyD, F87g, SZuJ, gZmw**) acknowledged that SelfOrg's empirical performance is strong and robust.

---

### **Addressed concerns:**

The main concerns raised by the reviewers are summarized below:

- **Hyperparameter Robustness [TXyD, SZuJ, gZmw].** In the rebuttal, we showed that SelfOrg uses one global cosine threshold $\tau$, fixed across 8 (+2 new) benchmarks and 10 backbone LLM models, and that $\tau$ is not critical for correctness: removing $\tau$ leaves accuracy essentially unchanged while slightly increasing token usage. Similarly, the early-stopping consensus parameter is only present in the efficient variant, serving to enhance token usage with comparable accuracy. **`[This concern was explicitly confirmed resolved by reviewers SZuJ and gZmw]`**

- **Assumption Validation [TXyD, F87g, SZuJ, gZmw].** We extended the validation to both open-ended and multiple-choice tasks, further validating our assumption that correct responses cluster, while incorrect ones are dispersed, holds in practice. We also related this to standard ensemble theory, explaining that meaningful collaboration requires agents to perform better than random. **`[This concern was explicitly confirmed resolved by reviewers SZuJ and gZmw]`**

- **Aggregation Method [TXyD, SZuJ].** We implemented several alternative aggregation methods, including those suggested by the reviewers, and observed that all perform similarly, suggesting that the primary performance driver is DAG-mediated message propagation, rather than the specific aggregation rule. **`[This concern was confirmed resolved by reviewer SZuJ]`**

### **Other addressed clarifications and additional experiments included in the rebuttal:**

To address reviewers’ remaining questions, we added the following analyses and experiments:

- scalability to a larger number of agents (20 and 50 agents) (addressing F87g),
- additional evaluations on HumanEval and SRDD collaborative coding (addressing F87g, gZmw),
- MAS-GPT (F87g) and self-consistency baselines (gZmw),
- heterogeneous agents with dynamic roles (F87g),
- memory + tool ablations (gZmw),
- multi-run results with standard deviations (F87g).

In addition, we responded to several smaller questions:
- clarified the meaning of “lightweight” (training-free orchestration, not lowest tokens) (TXyD),
- clarified why certain methods mentioned by the reviewer are not comparable baselines (F87g),
- clarified the basis for the “self-organization” claim (gZmw),
- provided detailed early-stopping breakdowns for Efficient SelfOrg (TXyD),
- added brief explanations where misunderstandings occurred (F87g).

Importantly, these additions were specifically acknowledged by reviewers SZuJ and gZmw during the discussion phase, both of whom stated that **our rebuttal had resolved their concerns** and that **they would advocate for acceptance.**

For the remaining reviewers, although they, unfortunately, could not participate in the discussion, **their concerns overlapped with those of SZuJ and gZmw, and those shared issues were explicitly validated as resolved.** The remaining questions were mostly clarifying or minor in scope, reflecting small misunderstandings; each was resolved through direct explanation or the requested experiments.

---

We would like to thank you all for reviewing our work and for the constructive feedback that strengthened our paper. We sincerely thank all reviewers: TXyD, F87g, SZuJ, and gZmw for their valuable time and thoughtful assessments, and extend special appreciation to reviewers SZuJ & gZmw for their participation in the discussion phase.

---

### Meta-Review · Area_Chair_9Tzk · 2026-01-02

**Summary:**

The manuscript proposes a response-conditioned framework that adapts communication on-the-fly. Agents
independently generate responses to the user query and assess peer contributions using an approximation of the Shapley value. They argue that the proposed framework, SelfOrg, enables the self-organization of agents without additional supervision or training.

While the idea is interesting, it has already been explored in the multi-agent systems literature [1, 2]; however, the manuscript fails to address/compare with these existing papers.

[1] Agentnet: Decentralized evolutionary coordination for llm-based multi-agent systems. NeurIPS 2025.
[2] MorphAgent: Empowering Agents through Self-Evolving Profiles and Decentralized Collaboration. Arxiv 2024

**Reviewer Concerns:**

## Reviewer TXyD
Reviewer TXyD gave the paper an initial rating of 4 (marginally below the acceptance threshold).

* Original key concerns:
    * hyperparameter sensitivity, resource usage and "lightweight" claim, insufficient evidence for key assumptions.
* Response to the concerns:
    * The authors argue the hyperparameter insensitivity through a fixed threshold across 8 benchmarks and 10 backbone models. The AC agrees with these statements.
    * The authors clarified that their use of "lightweight" refers to the orchestration cost (no training, no external judge LLM, no RL over edges), rather than the absolute lowest token count. The AC thinks it is reasonable.

## Reviewer F87g
Reviewer F87g raised several technical and evaluative concerns:
* Limited task diversity: the authors added new evaluations on coding tasks (HumanEval and SRDD collaborative coding).
* Missing baselines: the authors disagreed with the point that MAS-GPT is the most similar approach to SelfOrg.
* No statistical analysis: the authors have added multiple‑run results with standard deviations for the new coding tasks.
* Scalability: the authors ran a study with 20 and 50 agents. The AC believes it is insufficient to justify the scalability.

## Reviewer SZuJ
* The reviewer questioned whether the core assumption—that correct answers cluster while incorrect ones disperse—would hold for tasks with multiple equivalent answer forms. The authors clarified that because they use cosine similarity in the embedding space, paraphrases and equivalent phrasings naturally cluster together. They provided new correlation analyses across GSM-Hard, AQUA-RAT, and MMLU to further justify their statement.
* Reviewer SZuJ also questioned the hyperparameter insensitivity regarding the \tau (cosine-similarity threshold) and k (number of neighbors). The authors provided experimental data showing that even when filtering (removing $\tau$ and $k$) is omitted entirely, accuracy remains on par, while total token consumption only increases slightly.

## Reviewer gZmw
The reviewer gZmw again argued that the assumption "correct answers cluster while wrong ones scatter" only holds when the task difficulty is manageable for the model. The authors clarified that SELFORG is not claimed to "correct an incompetent backbone" but rather to coordinate capable backbones efficiently.

The authors also properly reply to the concerns of hyperparameter sensitivity and provide more baseline comparisons.

**Reviewer Scores:**

Reviewer TXyD and Reviewer F87g may increase their score from 4 to 6. However, AC still keeps the concern about the missing comparison with AgentNet.

---

### Decision · Program_Chairs · 2026-01-26

Accept (Poster)